# Discrepancies between pre-specified and reported primary outcomes: A cross-sectional analysis of randomized controlled trials in gastroenterology and hepatology journals

**Bing-Han Shang**[1], **Fang-Hui Yang**[1], **Yao Lin**[1], **Szymon Bialka**[2], **Dina Christa Janse van Rensburg**[3], **Adriano R. Tonelli**[4], **Sheikh Mohammed Shariful Islam**[5], **Izumi Kawagoe**[6], **Caroline Rhéaume**[7], **Kai-Ping Zhang**[1,8]\*

**1** Editorial Office, AME Publishing Company, Hong Kong, China, **2** Department of Anaesthesiology and Intensive Care, Faculty of Medical Sciences in Zabrze, Medical University of Silesia, Katowice, Poland, **3** Section Sports Medicine, Faculty of Health Sciences, University of Pretoria, Pretoria, South Africa, **4** Department of Pulmonary, Allergy and Critical Care Medicine, Respiratory Institute, Cleveland Clinic, Cleveland, OH, United States of America, **5** Institute for Physical Activity and Nutrition, School of Exercise and Nutrition Sciences, Deakin University, Geelong, Australia, **6** Department of Anesthesiology and Pain Medicine, Juntendo University School of Medicine, Tokyo, Japan, **7** Department of Family Medicine and Emergency Medicine, Faculty of Medicine, Université Laval, Québec, Canada, **8** Clinical Research Institute, Medical College, Nantong University, Nantong, China

\* zhangkp@amegroups.com

## Abstract

### Background

Previous research has raised concerns regarding inconsistencies between reported and pre-specified outcomes in randomized controlled trials (RCTs) across various biomedical disciplines. However, studies examining whether similar discrepancies exist in RCTs focusing on gastrointestinal and liver diseases are limited. This study aimed to assess the extent of discrepancies between registered and published primary outcomes in RCTs featured in journals specializing in gastroenterology and hepatology.

### Methods

We retrospectively retrieved RCTs published between January 1, 2017 and December 31, 2021 in the top three journals from each quartile ranking of the 2020 Journal Citation Reports within the "Gastroenterology and Hepatology" subcategory. We extracted data on trial characteristics, registration details, and pre-specified versus published primary outcomes. Pre-specified primary outcomes were retrieved from the World Health Organization's International Clinical Trials Registry Platform. Only trials reporting specific primary outcomes were included in analyzing primary outcome discrepancies. We also assessed whether there was a potential reporting bias that deemed to favor statistically significant outcomes. Statistical analyses included chi-square tests, Fisher's exact tests, univariate analyses, and logistic regression.

**Data Availability Statement:** All relevant data are within the paper and its supporting information files.

**Funding:** The author(s) received no specific funding for this work.

**Competing interests:** I have read the journal's policy and the authors of this manuscript have the following competing interests: Adriano R. Tonelli plays as an advisory board for Janssen and Merck and received grant from Janssen; Szymon Bialka is the secretary of the Silesian Branch of the Polish Society of Anesthesiology and Intensive Care from 2021 till now, the president of the Section of Regional Anesthesia and Pain Therapy and Polish Society of Anesthesiology and Intensive Care from 2021 til now, continued Treasurer of the Polish Society of Regional Anesthesia and Pain Therapy from 2023 till now, a co-creator patent device for isolating a patient with suspected infectious disease (exclusive right number: Pat.243051).

## Results

Of 362 articles identified, 312 (86.2%) were registered, and 79.8% of the registrations (249 out of 312) were prospective. Among the 285 trials reporting primary outcomes, 76 (26.7%) exhibited at least one discrepancy between registered and published primary outcomes. The most common discrepancies included different assessment times for the primary outcome (n = 32, 42.1%), omitting the registered primary outcome in publications (n = 21, 27.6%), and reporting the registered secondary outcomes as primary outcomes (n = 13, 17.1%). Univariate analyses revealed that primary outcome discrepancies were lower in the publication year 2020 compared to year 2021 (OR = 0.267, 95% CI: 0.101, 0.706, p = 0.008). Among the 76 studies with primary outcome discrepancies, 20 (26.3%) studies were retrospectively registered, and 32 (57.1%) of the prospectively registered trials with primary outcome discrepancies showed statistically significant results. However, no significant differences were found between journal quartiles regarding primary outcome consistency and potential reporting bias (p = 0.14 and p = 0.28, respectively).

## Conclusions

This study highlights the disparities between registered and published primary outcomes in RCTs within gastroenterology and hepatology journals. Attention to factors such as the timing of primary outcome assessments in published trials and the consistency between registered and published primary outcomes is crucial. Enhanced scrutiny from journal editors and peer reviewers during the review process is necessary to ensure the reliability of gastrointestinal and hepatic trials.

## Introduction

Gastroenterology and hepatology are fields that often involve multifaceted treatment regimens and diverse patient populations. In 2019, digestive diseases accounted for more than one-third of prevalent disease cases, representing a significant global health care burden [1]. In the era of evidence-based medicine, high-quality randomized controlled trials (RCTs) stand as pivotal sources of evidence in scientific research, owing to their robust study designs and significant value [2]. These trials often serve as primary references for formulating clinical guidelines and shaping medical decision-making. However, numerous trials encounter the issue of selective and incomplete reporting of results, which distorts their evidence-based value [3–6]. The Centre for Evidence-Based Medicine Outcome Monitoring Project has found that, on average, each trial in top-ranked medical journals silently adds 5.3 new outcomes [7]. Such discrepancies in trial results can potentially exaggerate benefits or underestimate adverse outcomes, leading to misguided clinical recommendations, wastage of resources, and, in severe cases, harm to patients [8].

To address these concerns, prospective registration of RCTs becomes imperative to ensure transparency and complete disclosure of proposed primary outcomes. Prospective research protocols and registrations are critical in curbing incomplete or selective reporting by serving as predetermined blueprints for evaluating comprehensive reports and facilitating comparisons. Recognizing this need, the International Committee of Medical Journal Editors (ICMJE) announced in 2004 that prospective registration of clinical trials in a public trial registry would

be a prerequisite for publication consideration starting in 2005 [9]. This proactive initiative aims to reinforce transparency and alleviate reporting bias by enabling comparisons between the outcomes initially planned for the trial and those ultimately reported.

Despite strides toward improving timely trial registration, a significant number of published trials remained unregistered or lacked prospective registration altogether [8, 10–12]. A study examining RCTs registered in any World Health Organization trial registry platform in 2018 found that only 41.7% were registered prospectively [8]. Moreover, selective outcome reporting bias persists across biomedical disciplines, including anesthesiology, psychology, otorhinolaryngology, headache medicine, mental health and orthopaedical surgery despite the implementation of ICMJE guidelines. Studies in these biomedical fields have reported a huge field difference in discrepancies between registered and published primary outcomes, ranging from 25.9% to 92% [13–18].

Despite the huge field difference, research on this important topic remains limited in the field of gastroenterology and hepatology. A single relevant study by Li et al. revealed a discrepancy rate of 14.2% between registered and reported results in five general and internal medicine journals and five gastroenterology and hepatology journals with the highest impact factors in the 2011 Clarivate Analytics Journal Citation Report (JCR) report [19]. However, the study did not compare discrepancies based on journal quartiles. It is also crucial to assess whether the situation has improved over the past decade. Therefore, a comprehensive and updated evaluation of the consistency between registered and published primary outcomes is warranted within the context of gastrointestinal and hepatic journals.

Therefore, in this study, we aimed to: 1) analyze the distribution of RCT registration from 2017 to 2021 and assess variations in registration practices among gastrointestinal and hepatic journals; 2) compare the primary outcomes initially pre-specified during registration with the final outcomes reported in subsequent publications, aiming to identify the extent of reporting bias, and evaluate whether any bias tends towards reporting significant results; and 3) investigate the factors influencing the inconsistency between the registration and publication of primary outcomes.

## Materials & methods

### Search strategy and eligibility criteria

In this cross-sectional analysis, we retrospectively selected the top three journals from each quartile within the "Gastroenterology and Hepatology" subcategory of the 2020 JCR. Journals that did not include the original article, as per their author instructions and our preliminary searches, were excluded. Primary reports of RCTs published in the 12 chosen gastrointestinal and hepatic journals over a 5-year period (January 1, 2017 to December 31, 2021) were identified using PubMed as of March 2022. The details of the final search are provided in **S1 File**. We present this article in accordance with the STrengthening the Reporting of OBservational studies in Epidemiology (STROBE) reporting checklist (**S2 File**).

We employed the Cochrane Handbook's definition of RCT, which characterizes it as "A clinical trial that involves at least one test treatment and one control treatment, concurrent enrollment and follow-up of the test- and control-treated groups, and in which the treatments to be administered are selected by a random process, such as the use of a random numbers table" [20]. The inclusion criteria are RCTs published in English within the 12 eligible journals that specifically assessed the intervention in human subjects. We excluded reviews, observational studies, systematic reviews or meta-analyses, cost-effectiveness analyses, animal or *in vitro* studies, case reports, cross-over studies, as well as ancillary studies (e.g., protocol studies, secondary analyses, follow-up studies, subgroup analyses, and post hoc analyses). We accessed

free articles directly through open-access journals. For articles that were not freely available, we thoroughly searched the full text of each article using multiple sources, including Web of Science, Embase, and Scopus, among others. Articles for which full text was unattainable through these means were subsequently excluded from our analysis.

## Study selection and data extraction

The reports obtained from the search were imported into Endnote (version 20; Clarivate Analytics, USA). A total of three investigators (BHS, FHY, and YL) formed cross-groups in pairs, i.e., three groups. Two investigators from each group independently screened the articles for eligibility, initially by title and abstract, followed by a full-text assessment. Any disagreements were resolved through discussion until a consensus was reached.

We developed a standardized extraction form using Microsoft Excel (2022 version; Microsoft Corporation, USA) to mitigate potential bias, encompassing journal information, article characteristics, registry details, and primary outcome discrepancies (S3 File). All investigators underwent training to ensure consistency and minimize discrepancies during data extraction. For article characteristics, one investigator (YL) reviewed the full text of each study and extracted the data, which were then cross-validated by another investigator (BHS). Any conflicts were resolved through discussion with a third reviewer (FHY) until a consensus was reached. For registry information, we utilized the World Health Organization's International Clinical Trials Registry Platform (ICTRP), comprising 20 main trial registries globally, to identify and download registration records and to retrieve pre-specified primary outcomes of published trials. This approach ensured uniformity in search mechanisms. If no registration number was provided in the publication, we manually searched the ICTRP using the publication title, author names, trial participants, and primary sponsors to identify any possible registration numbers. If no registry number was uncovered in this way, the trial was deemed unregistered. If a registration number was provided in the publication, we entered the registration number into the ICTRP to retrieve relevant registration information. If the authors provided a registration number but we did not find a corresponding registration record, we treated these trials in the same manner as studies that did not report a registration number at all.

Primary outcomes were defined as those explicitly reported in the study. In cases where no outcome measure was explicitly named as primary, we recorded the outcome stated in the sample size calculation of the study. Additionally, if neither was available, we adopted a conservative approach, categorizing the article as having no reported primary outcome and subsequently excluding it from the analysis of outcome discrepancies. Two investigators (BHS, FHY) compared the primary outcomes reported in the articles with those initially registered to determine consistency. These outcomes reflect those specified at the time of the initial registration and do not include any outcomes added or modified in subsequent updates. We chose these primary outcomes because, while the ICTRP provides a comprehensive registry, it does not always allow for a clear assessment of historical versions or the trial stage at which changes were made. For instance, prospective registrations may have retrospective changes where authors modify primary outcomes after concluding the trial. We extracted data from 10 random samples of RCTs to ensure a consistent understanding of discrepancies. Each discrepancy was independently categorized into five types, based on criteria initially proposed by Chan et al. and refined by Mathieu et al. [5, 21]:

1. The registered primary outcome was reported as a secondary outcome in the published article.

2. The registered primary outcome was omitted in the published report.

3. A new primary outcome was introduced in the published article (i.e., an outcome that does not appear at all in the registry but is introduced as primary in the article).

4. The published primary outcome was described as a secondary outcome in the registry.

5. The timing of the assessment of primary outcomes differed between the registered and published data.

Similar to Chan et al.'s study [5], a discrepancy was deemed to favor statistically significant outcomes "if a new statistically significant primary outcome was introduced in the published articles, or if a nonsignificant primary outcome was omitted (e.g., the omitted outcome might not have achieved statistical significance, leading to its exclusion from the published results. This assumption is based on the notion that statistically significant results are more likely to be reported and published due to the well-known publication bias) or defined as nonprimary in the published articles, or if registered statistically significant secondary outcomes became published primary outcomes". Articles retrospectively registered or lacking explicit primary outcomes were excluded from this analysis, as it was not feasible to ascertain whether any modifications to the primary outcome had occurred.

## Statistical analysis

Statistical analyses are conducted using Microsoft Excel (2022 version; Microsoft Corporation, USA) and IBM SPSS Statistics (version 22; IBM, Armonk, NY, USA). Descriptive statistical analyses are employed to outline the basic characteristics of included studies within the four quartiles. In addition, the proportion of trials exhibiting inconsistent primary outcome measures between registration and publication, as well as the proportion reporting favorable results, are determined and stratified by quartiles. For continuous variables, mean and standard deviation (for normally distributed data) or the median and range (for non-normally distributed data) are reported. Categorical variables are presented as frequencies and percentages.

The chi-square test or Fisher's exact test is used to assess differences in each trial characteristic across domains, including registration status and registration type. We also used these tests to assess primary outcome consistency differences and report favorable results across journal domains. Additionally, univariate analyses are performed to determine the effect of each variable on primary outcome discrepancies. Multivariable logistic regression includes all the significant variables in the univariable analysis to identify factors influencing discrepancies between the registered and reported primary outcomes. A significance level of $p < 0.05$ (two sides) is applied to all analyses. Cohen's kappa coefficient is used to assess inter-observer variability in judging discrepancies. Cohen's kappa coefficient measures the agreement of evaluators, with values interpreted as follows: $\leq 0.2$ indicating poor agreement, 0.21–0.40 fair agreement, 0.41–0.60 moderate agreement, 0.61–0.80 strong agreement, and $\geq 0.81$ very good agreement [22].

## Results

Twelve journals within the gastrointestinal and hepatic domain were identified from the 2020 JCR report, distributed across four quartiles. However, *Nature Reviews Gastroenterology & Hepatology* in Quartile 1 and *Gastroenterology Clinics of North America* in Quartile 3 did not include original articles. After that, we excluded these two journals and included another two sequentially (*Gastroenterology* and *Techniques in Coloproctology*). The eligible journals in the specialty of "Gastroenterology and Hepatology" are as follows: Quartile 1 (*Journal of Hepatology*, *GUT*, and *Gastroenterology*), Quartile 2 (*Hepatology International*, *Liver International*, and *Liver Transplantation*), Quartile 3 (*Expert Review of Gastroenterology & Hepatology*,

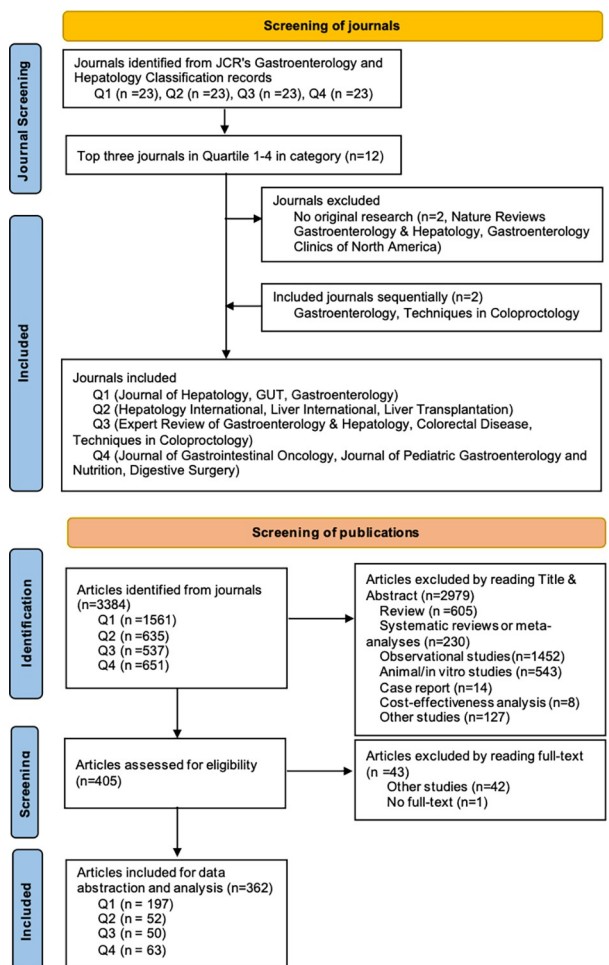

**Fig 1. Flow diagram of identification of journals and articles.**

*Colorectal Disease*, and *Techniques in Coloproctology*), Quartile 4 (*Journal of Gastrointestinal Oncology, Journal of Pediatric Gastroenterology and Nutrition*, and *Digestive Surgery*) (**Fig 1**). Mandatory registration of RCTs was not found in *GUT* (Quartile 1), and *Hepatology International* (Quartile 1). Among the trials published in journals requiring mandatory registration, 14.5% (41/282) were not registered (**Table 1**).

## Search results and trial characteristics

A total of 3,384 records were screened. **Fig 1** shows the publication selection process. Exclusion reasons included review type (n = 605), systematic reviews or meta-analyses (n = 230), observational studies (n = 1,452), animal or *in vitro* studies (n = 543), case reports (n = 14), cost-effectiveness analysis (n = 8), and other studies (n = 127). Subsequently, 405 articles underwent full-text screening for eligibility. Of them, 42 cross-over studies and ancillary studies were further excluded. And we were unable to obtain the full text of one trial. Finally, 362 trials were eligible for inclusion.

**Table 2** presents the characteristics of the 362 eligible trials. Of note, the highest number of RCTs was published in Quartile 1, accounting for 54.4% (197/362). Sample sizes ranged widely, with a median of 115 and a range spanning from 12 to 32,063. Overall, larger trial

**Table 1. Editorial policies and proportion of trials of included journals.**

| Rank based on impact factor | Journal | 2020 IF | ICMJE member or adherence to ICMJE guidelines | CONSORT endorsed | Mandatory RCT registration[#] | No. Included RCTs | Mandatory but unregistered trials, n (%) |
|---|---|---|---|---|---|---|---|
| Q1 | *Journal of Hepatology* | 25.083 | Yes | Yes | "Trials must register at or before the onset of patient enrolment" | 34 | 3(8.8) |
| | *GUT* | 23.059 | Yes | Yes | No | 61 | 2(3.3) * |
| | *Gastroenterology* | 22.682 | Yes | Yes | "Authors of manuscripts involving clinical trials must provide full registration of their trial(s)" | 102 | 7(6.9) |
| Q2 | *Hepatology International* | 6.047 | Yes | Not indicated | No | 19 | 7(36.8) * |
| | *Liver International* | 5.828 | Yes | Yes | "Authors are asked to include the name of the trial register and the clinical trial registration number" | 22 | 3(13.6) |
| | *Liver Transplantation* | 5.799 | Yes | Yes | "Trials must be registered in a registry" | 11 | 3(27.3) |
| Q3 | *Expert Review of Gastroenterology & Hepatology* | 3.869 | Yes | Not indicated | "All clinical trials must have been registered in a public repository" | 2 | 1(50.0) |
| | *Colorectal Disease* | 3.788 | Yes | Yes | "These must have been registered in an international public registry of controlled trials before submission to the journal" | 34 | 3(8.8) |
| | *Techniques in Coloproctology* | 3.781 | Yes | Yes | "Authors must register prospective clinical trials (phase II to IV trials) in suitable publicly available repositories" | 14 | 7(50.0) |
| Q4 | *Journal of Gastrointestinal Oncology* | 2.892 | Yes | Yes | "The study project registration number (e.g., registration number for a clinical trial, . . .) should be included at the end of the abstract" | 8 | 5(62.5) |
| | *Journal of Pediatric Gastroenterology and Nutrition* | 2.839 | Yes | Yes | "All trials submitted to the Journal with patient enrollment commencing after January 1, 2009 must be registered in a public trials registry" | 49 | 5(10.2) |
| | *Digestive Surgery* | 2.588 | Yes | Yes | "If your manuscript is a clinical trial, please provide the clinical trial number" | 6 | 4(66.7) |

Q: Quartile; IF: Impact Factor; ICMJE: International Committee of Medical Journal Editors; CONSORT: Consolidated Standards of Reporting Trials; RCT: randomized controlled trials.

[#]: The journal requirements are retrieved up to March 8th, 2024

*: The data represent the number and percentage of trials without registration among journals that do not require a mandatory registration.

sample sizes were observed in Quartile 1 journals compared to Quartiles 2 to 4. The unspecified clinical phase of the study (50.0%), multicenter institutions (57.2%), nonprofit-funded (43.1%), superiority trial design (93.4%), efficiency/tolerance/safety outcome (98.9%), and European study site (40.9%) predominated among the publications in these trials. Only 61 (16.9%) studies declared adherence to the Consolidated Standards of Reporting Trials (CONSORT) reporting checklist in the main text.

## Registration

Table 3a and 3b show the registration and prospective registration by year of publication, study location, quartile, funding, and trial design, respectively. Among the included RCTs, 86.2% (312/362) were registered (Table 3a), with 79.8% (249/312) registered prospectively and 20.2% (63/312) registered retrospectively (Table 3b). The registration rate of RCT studies had

**Table 2. Basic characteristics of the included RCT studies.**

| Characteristic | Category | Total (n = 362) | Q1 (n = 197) | Q2 (n = 52) | Q3 (n = 50) | Q4 (n = 63) |
|---|---|---|---|---|---|---|
| **Sample size, median (range)** | | 115(12–32,063) | 159(12–32,063) | 117(22–596) | 70.5(15–512) | 80(24–633) |
| **Phase of study, n (%)** | I | 13(3.6) | 8(4.1) | 1(1.9) | 0(0.0) | 4(6.3) |
| | II | 71(19.6) | 52(26.4) | 12(23.1) | 4(8.0) | 3(4.8) |
| | III | 67(18.5) | 43(21.8) | 8(15.4) | 9(18.0) | 7(11.1) |
| | IV | 30(8.3) | 16(8.1) | 7(13.5) | 3(6.0) | 4(6.3) |
| | Not specified | 181(50.0) | 78(39.6) | 24(46.2) | 34(68.0) | 45(71.4) |
| **Study centers, n (%)** | Single center | 103(28.5) | 28(14.2) | 13(25.0) | 23(46.0) | 39(61.9) |
| | Multicenter | 207(57.2) | 148(75.1) | 26(50.0) | 17(34.0) | 16(25.4) |
| | Not specified | 52(14.4) | 21(10.7) | 13(25.0) | 10(20.0) | 8(12.7) |
| **Funding, n (%)** | Industry | 109(30.1) | 78(39.6) | 14(26.9) | 7(14.0) | 10(15.9) |
| | Nonprofit | 156(43.1) | 93(47.2) | 19(36.5) | 20(40.0) | 24(38.1) |
| | Mixed | 17(4.7) | 9(4.6) | 4(7.7) | 2(4.0) | 2(3.2) |
| | None disclosed/no funding | 80(22.1) | 17(8.6) | 15(28.8) | 21(42.0) | 27(42.9) |
| **Trial design, n (%)** | Superiority | 338(93.4) | 181(91.9) | 48(92.3) | 49(98.0) | 60(95.3) |
| | Equivalency | 4(1.1) | 0(0.0) | 3(5.8) | 0(0.0) | 0(0.0) |
| | Noninferiority | 20(5.5) | 16(8.1) | 1(1.9) | 1(2.0) | 3(4.8) |
| **Study outcome, n (%)** | Efficiency/tolerance/safety | 358(98.9) | 193(98.0) | 52(100.0) | 50(100.0) | 63(100.0) |
| | Toxicity/harm | 4(1.1) | 4(2.0) | 0(0.0) | 0(0.0) | 0(0.0) |
| **Registration, n (%)** | Yes | 312(86.2) | 185(93.9) | 39(75.0) | 39(78.0) | 49(77.8) |
| | No | 50(13.8) | 12(6.1) | 13(25.0) | 11(22.0) | 14(22.2) |
| **Registration status, n (%)** | Prospective | 249(79.8) * | 163(88.1) | 25(64.1) | 31(79.5) | 30(61.2) |
| | Retrospective | 63(20.2) * | 22(11.9) | 14(35.9) | 8(20.5) | 19(38.8) |
| **Study location, n (%)** | Europe | 148(40.9) | 85(43.1) | 13(25.0) | 29(58.0) | 21(33.3) |
| | America | 100(27.6) | 65(33.0) | 11(21.2) | 3(6.0) | 21(33.3) |
| | Asia | 99(27.3) | 43(21.8) | 26(50.0) | 11(22.0) | 19(30.2) |
| | Oceania | 9(2.5) | 3(1.5) | 0(0.0) | 5(10.0) | 1(1.6) |
| | Africa | 6(1.7) | 1(0.5) | 2(3.8) | 2(4.0) | 1(1.6) |
| **CONSORT endorsed, n (%)** | Yes | 61(16.9) | 28(14.2) | 5(9.6) | 14(28.0) | 14(22.2) |
| | No | 301(83.1) | 169(85.8) | 47(90.4) | 36(72.0) | 49(77.8) |
| **Mandatory RCT registration, n (%)** | Yes | 282(77.9) | 136(69.0) | 33(63.5) | 50(100.0) | 63(100.0) |
| | No | 80(22.1) | 61(31.0) | 19(36.5) | 0(0.0) | 0(0.0) |

Q: Quartile; CONSORT: Consolidated Standards of Reporting Trials; RCT: randomized controlled trials.

*: The portion of prospective or retrospective registration of the registered study (n = 312).

no significant difference over the five-year period (p = 0.83) (**Table 3a**). However, there were significant differences in study location, quartile, and funding between registered and unregistered trials (p = 0.009, p<0.001, and p = 0.01, respectively, **Table 3a**). Furthermore, there were significant differences between prospective and retrospective registration in quartile, funding, and trial design (p<0.001, p = 0.001, p = 0.006, respectively) (**Table 3b**).

## Discrepancy of registered and published primary outcomes

In analyzing discrepancies between registered and published primary outcomes, Cohen's kappa coefficient among the three reviewers for the extraction of differences was 0.872, indicating very good agreement. Only studies reporting primary outcomes were included in further analyses. Among the 312 registered trials, 285 were eligible for primary outcome

**Table 3. Registration and prospective registration across clinical trials, stratified by trial characteristics.**

| Table 3a | | | | |
|---|---|---|---|---|
| **Feature** | **Total (n = 362)** | **Registered trials (n = 312)** | **Unregistered trials(n = 50)** | **P-Value** |
| **Year of publication, n (%)** | | | | 0.83 |
| 2017 | 73 | 63(86.3) | 10(13.7) | |
| 2018 | 64 | 52(81.3) | 12(18.7) | |
| 2019 | 82 | 76(92.7) | 6(7.3) | |
| 2020 | 75 | 61(81.3) | 14(18.7) | |
| 2021 | 68 | 60(88.2) | 8(11.8) | |
| **Study Location, n (%)** | | | | |
| Europe | 148 | 131(88.5) | 17(11.5) | 0.009 * |
| America | 100 | 91(91.0) | 9(9.0) | |
| Asia | 99 | 78(78.8) | 21(21.2) | |
| Oceania | 9 | 9(100.0) | 0(0.0) | |
| Africa | 6 | 3(50.0) | 3(50.0) | |
| **Quartile, n (%)** | | | | <0.001* |
| Q1 | 197 | 185(93.9) | 12(6.1) | |
| Q2 | 52 | 39(75.0) | 13(25.0) | |
| Q3 | 50 | 39(78.0) | 11(22.0) | |
| Q4 | 63 | 49(77.8) | 14(22.2) | |
| **Funding, n (%)** | | | | 0.01* |
| Industry | 109 | 97(89.0) | 12(11.0) | |
| Nonprofit | 156 | 139(89.1) | 17(10.9) | |
| Mixed | 17 | 16(94.1) | 1(5.9) | |
| None disclosed/no funding | 80 | 60(75.0) | 20(25.0) | |
| **Trial design, n (%)** | | | | 0.21 |
| Superiority | 338 | 289(85.5) | 49(14.5) | |
| Equivalency | 4 | 4(100.0) | 0(0.0) | |
| Noninferiority | 20 | 19(95.0) | 1(5.0) | |
| Table 3b | | | | |
| **Feature** | **Total (n = 312)** | **Prospective Registration (n = 249)** | **Retrospective Registration (n = 63)** | **P-value** |
| **Year of publication, n (%)** | | | | 0.92 |
| 2017 | 63 | 51(81.0) | 12(19.0) | |
| 2018 | 52 | 42(80.8) | 10(19.2) | |
| 2019 | 76 | 57(75.0) | 19(25.0) | |
| 2020 | 61 | 53(86.9) | 8(13.1) | |
| 2021 | 60 | 46(76.7) | 14(23.3) | |
| **Study Location, n (%)** | | | | 0.74 |
| Europe | 131 | 105(80.2) | 26(19.8) | |
| America | 91 | 76(83.5) | 15(16.5) | |
| Asia | 78 | 59(75.6) | 19(24.4) | |
| Oceania | 9 | 7(77.8) | 2(22.2) | |
| Africa | 3 | 2(66.7) | 1(33.3) | |
| **Quartile, n (%)** | | | | <0.001* |
| Q1 | 185 | 163(88.1) | 22(11.9) | |
| Q2 | 39 | 25(64.1) | 14(35.9) | |

(*Continued*)

**Table 3.** (Continued)

| | | | | |
|---|---|---|---|---|
| Q3 | 39 | 31(79.5) | 8(20.5) | |
| Q4 | 49 | 30(61.2) | 19(38.8) | |
| **Funding, n (%)** | | | | 0.001* |
| Industry | 97 | 90(92.8) | 7(7.2) | |
| Nonprofit | 139 | 102(73.4) | 37(26.6) | |
| Mixed | 16 | 14(87.5) | 2(12.5) | |
| None disclosed/no funding | 60 | 43(71.7) | 17(28.3) | |
| **Trial design, n (%)** | | | | 0.006* |
| Superiority | 289 | 230(79.6) | 59(20.4) | |
| Equivalency | 4 | 1(25.0) | 3(75.0) | |
| Noninferiority | 19 | 18(94.7) | 1(5.3) | |

Q: Quartile.

*: P< 0.05.

discrepancy analysis, as 27 (8.7%) RCTs were excluded due to imprecise primary outcomes. A total of 26.7% (76/285) of the trials exhibited at least one major discrepancy between the registry and publication of the primary outcomes (**Table 4**). As detailed in **Table 4**, the most common discrepancy was different assessment times for the registered versus published primary outcomes (n = 32, 42.1%). Other notable discrepancies included registered primary outcomes omitted in publications (n = 21, 27.6%), registered secondary outcomes promoted to primary outcomes (n = 13, 17.1%), registered primary outcomes demoted to secondary outcomes (n = 6, 7.9%), and new primary outcomes introduced (n = 4, 5.3%). No significant differences were found in primary outcome inconsistencies across different journal quartiles (p = 0.14). Univariate analyses revealed that only the publication year (year 2020) was associated with the discrepancy between registered and published primary outcomes (OR = 0.267, 95% CI:0.101,0.706, p = 0.008) (**Table 5**); therefore, multivariable logistic regression analysis was not conducted. Specifically, the result indicated that in 2020, the likelihood of primary

**Table 4. Frequency of major discrepancies between registry and publication, by different journal quartiles.**

| | Total No. (%) | Q1 | Q2 | Q3 | Q4 | P-value |
|---|---|---|---|---|---|---|
| **Discrepancy in published articles relative to registered trials** | (n = 76) | (n = 40) | (n = 13) | (n = 9) | (n = 14) | 0.14 |
| The registered primary outcome was reported as a secondary outcome in the published article | 6(7.9) | 3 (7.5) * | 0(0.0) | 1(11.1) | 2(14.3) | |
| The registered primary outcome was omitted in the published report. | 21(27.6) | 15(37.5) * | 1(7.7) | 2(22.2) * | 3(21.4) * | |
| A new primary outcome was introduced in the published article | 4(5.3) | 2(5.0) | 0(0.0) | 2(22.2) | 0(0.0) | |
| The published primary outcome was described as a secondary outcome in the registry. | 13(17.1) | 7(17.5) | 4(30.8) | 0(0.0) | 2(14.3) | |
| The timing of assessment of the registered and published primary outcomes differed | 32(42.1) | 13(32.5) | 8(61.5) | 4(44.4) | 7(50.0) | |
| **Discrepancy of primary outcomes favoring significant results** | (n = 32) | (n = 21) | (n = 4) | (n = 4) | (n = 3) | 0.28 |
| A nonsignificant primary outcome was omitted or defined as nonprimary in the published articles | 15(46.9) | 12(57.2) | 1(12.5) | 1(25.0) | 1(33.3) | |
| Registered statistically significant secondary outcomes became published primary outcomes | 4(12.5) | 2(9.5) | 0(0.0) | 2(50.0) | 0(0.0) | |
| A new statistically significant primary outcome was introduced in the published articles | 13(40.6) | 7(33.3) | 3(37.5) | 1(25.0) | 2(66.6) | |

Q: Quartile.

*: Articles involved two types of discrepancies.

**Table 5. Univariate analyses of factors related to a discrepancy of the primary outcome in the publication.**

| Characteristic | Total, N | Primary-Outcome agreement trial, N (%) | Primary-Outcome disagreement trial, N (%) | Univariate analyses | |
|---|---|---|---|---|---|
| | N = 285 | N = 209 | N = 76 | Odds ratio (95% CI) | P-value |
| **Quartile** | | | | | |
| Q4 [a] | 41 | 27(65.9) | 14(34.1) | | |
| Q1 | 174 | 134(77.0) | 40(23.0) | 0.576 (0.276,1.202) | 0.14 |
| Q2 | 34 | 21(61.8) | 13(38.2) | 1.194 (0.464,3.075) | 0.71 |
| Q3 | 36 | 27(75.0) | 9(25.0) | 0.643 (0.238,1.735) | 0.38 |
| **Publication year** | | | | | |
| 2021 [a] | 53 | 35(66.0) | 18(34.0) | | |
| 2017 | 56 | 37(66.1) | 19(33.9) | 0.998 (0.452,2.207) | >0.99 |
| 2018 | 50 | 39(78.0) | 11(22.0) | 0.548 (0.228,1.320) | 0.18 |
| 2019 | 68 | 47(69.1) | 21(30.9) | 0.869 (0.404,1.870) | 0.72 |
| 2020 | 58 | 51(87.9) | 7(12.1) | 0.267 (0.101,0.706) | 0.008* |
| **Study location** | | | | | |
| Africa [a] | 3 | 2(66.7) | 1(33.3) | | |
| Europe | 120 | 85(70.8) | 35(29.2) | 0.824 (0.072,9.378) | 0.88 |
| America | 82 | 65(79.3) | 17(20.7) | 0.523(0.045,6.117) | 0.61 |
| Asia | 72 | 50(69.4) | 22(30.6) | 0.880 (0.076,10.221) | 0.92 |
| Oceania | 8 | 7(87.5) | 1(12.5) | 0.286 (0.012,6.914) | 0.44 |
| **Study center** | | | | | |
| Multicenter [a] | 177 | 133(75.1) | 44(24.9) | | |
| Single center | 74 | 51(68.9) | 23(31.1) | 1.088 (0.472,2.507) | 0.84 |
| Not specified | 34 | 25(73.5) | 9(26.5) | 1.363 (0.749,2.481) | 0.31 |
| **Phase of study** | | | | | |
| Not specified [a] | 130 | 90(69.2) | 40(30.8) | | |
| I | 10 | 8(80) | 2(20.0) | 0.563 (1.114,2.768) | 0.48 |
| II | 64 | 49(76.6) | 15(23.4) | 0.689(0.346,1.137) | 0.29 |
| III | 54 | 42(77.8) | 12(22.2) | 0.643 (0.306,1.350) | 0.24 |
| IV | 27 | 20(74.1) | 7(25.9) | 0.788 (0.308,2.012) | 0.62 |
| **Funding** | | | | | |
| Mixed [a] | 11 | 9(81.8) | 2(18.2) | | |
| None/NA | 52 | 36(69.2) | 16(30.8) | 2.000 (0.387,10.325) | 0.408 |
| Profit | 94 | 76(80.9) | 18(19.1) | 1.066 (0.212,5.364) | 0.938 |
| Non-profit | 128 | 88(68.8) | 40(31.2) | 2.045 (0.423,9.902) | 0.374 |
| **Trial design** | | | | | |
| Noninferiority [a] | 18 | 15(83.3) | 3(16.7) | | |
| Superiority | 264 | 191(72.3) | 73(27.7) | 1.911 (0.537,6.795) | 0.32 |
| Equivalency | 3 | 3(100.0) | 0(0) | 0 | >0.99 |
| **Registration status** | | | | | |
| Prospective [a] | 227 | 171(75.3) | 56(24.7) | | |
| Retrospective | 58 | 38(65.5) | 20(34.5) | 1.607 (0.865,2.987) | 0.13 |
| **CONSORT endorsed** | | | | | |
| Yes [a] | 47 | 37(78.7) | 10(21.3) | | |
| No | 238 | 172(72.3) | 66(27.7) | 1.420 (0.668,3.018) | 0.36 |
| **Mandatory RCT registration** | | | | | |
| Yes [a] | 217 | 158(72.8) | 59(27.2) | | |
| Not | 68 | 51(75) | 17(25.0) | 0.893(0.478,1.668) | 0.72 |

Q: Quartile; CONSORT: Consolidated Standards of Reporting Trials; RCT: randomized controlled trials; CI: confidence interval.

[a]: Reference category; *: P< 0.05

outcome discrepancy was 0.267 times that of 2021, and this difference was statistically significant. The proportion of studies with primary outcome discrepancies was lower in 2020 compared to 2021 (12.1% vs 34.0%). Journal quartile, study location, study center, phase of the study, funding, trial design, registration status, CONSORT endorsed, and whether RCT registration was mandatory were not associated with the discrepancy of registered and published primary outcomes.

### Discrepancy of primary outcomes favoring significant results

In the analysis of outcome reporting bias, 20 studies that were retrospectively registered among the 76 papers with a discrepancy between the registry and the publication were excluded. Among the remaining 56 studies, 57.1% (32/56) exhibited a discrepancy favoring a statistically significant primary outcome (**Table 4**). Specifically, 46.9% (15/32) of the discrepancies involved omitting non-significant registered primary outcomes, and 40.6% (13/32) involved the introduction of significant new primary outcomes in publications. No significant differences were observed in potential outcome reporting bias across journal quartiles (p = 0.28).

## Discussion

Our study reveals several key findings: 1) Not all 12 enrolled journals mandated trial registration, and over 10% of unregistered trials were still published in journals requiring registration. The registration rate of the 362 eligible RCTs had no significant difference over time from 2017 to 2021, with 86.2% (312 out of 362) registered and 79.8% (249 out of 312) registered predominantly prospectively. Both registration and prospective registration were statistically different by journal quartile and funding. 2) Over a quarter of the trials exhibited evidence of discrepancies in primary outcomes, with the top three discrepancies being differences in assessment times (42.1%), omission of primary outcomes (27.6%), and reporting the registered secondary outcomes as primary outcomes (17.1%). In addition, over half of the discrepancy favored a statically significant primary outcome, mainly attributed to the omission of non-significant registered primary outcomes and the introduction of significant new primary outcomes. 3) Primary outcome discrepancies were lower in the publication year 2020 compared to year 2021 (OR = 0.267, 95% CI: 0.101, 0.706, p = 0.008). However, no such associations were found regarding journal quartile, study location or center, funding, trial design, registration status, study phase, adherence to CONSORT, or registration compulsion. Furthermore, no significant differences were observed concerning potential reporting bias across journal quartiles.

### Comparison with similar research and explanations of findings

**Trial registration.** Adherence to prospective trial registration varies across medical fields, with studies in psychiatry, pediatric surgery, and anaesthesia reporting a range from 33.1% to 71.1% [14, 23, 24]. Compared to a previous investigation on reporting bias in gastroenterology [19], our research reveals a substantially higher proportion of prospectively registered studies (79.8% vs. 37.2%), indicating a remarkable improvement in the registration of RCT studies within this field. Nevertheless, 14.5% of studies were still published without registration in journals requiring mandatory registration. Our study identified variations in how journals, despite claiming adherence to ICMJE guidelines or being ICMJE members, articulate registration requirements in the author's instructions. Vague or ambiguous statements may contribute to a lack of rigorous enforcement by authors or journal editors. As reflected in our findings, mandatory registration requirements appear to be a significant motivator for registration.

When extracting data, we observed invalid trial registration numbers, indicating potential lapses in the meticulous verification of registration information by journal editors or reviewers. Factors associated with prospective registration, such as publication in high-impact journals, non-profit funding sources, and trial designs, present opportunities to promote early registration at different stages of the research process. A study in surgical journals [25] reported a notable association between trial registration and higher journal impact factors, aligning with our research findings. These insights underscore the need to continue enhancing and standardizing trial registration practices in this field.

**Discrepancy in primary outcomes.** Primary outcomes play a crucial role in assessing the effectiveness of interventions for specific symptoms or the disease of interest, providing the strongest evidence for their maximum effects [26, 27]. The pre-specification of primary outcomes is essential as it prevents deviations from study protocols and selective reporting. When pre-defined outcomes are altered or omitted, this protective mechanism is compromised. Our results align with previous research that found differences between registry entries and publications [14, 15, 28–31]. We observed a higher level of outcome reporting bias compared to an earlier gastroenterology study (26.7% vs 14.2%) [19], possibly because the previous analysis focused on the top 5 impact factor journals, while our study assessed research across all impact factor quartiles. This highlights that despite the increased registration rate of RCTs, the divergence in reporting main results has concurrently risen. Our study underscores the persistent challenge of outcome reporting bias in gastrointestinal research, emphasizing that registration alone does not eliminate the risk of selective outcome reporting. Notably, our investigation into the varied nature of discrepancies revealed that differences in the timing of primary outcome assessments accounted for the highest proportion of discrepancies in our study, which may often be overlooked or underappreciated. Different assessment times can impact the demonstration of efficacy or harm, leading to biased findings if earlier non-significant timepoints were registered while optimal timepoints were chosen to report significant effects. Additionally, omitting a primary outcome and reporting the registered secondary outcomes as primary outcomes were the second and third most common discrepancies. A point claimed that the discrepancies were because the researchers discovered unintended effects or harms caused by the intervention, leading to the selective omission of pre-specified primary outcomes [32]. However, it is important to note that changes to primary endpoints do not inherently indicate poor practice. There can be plausible reasons for such changes, which should be transparently discussed in the study publication. Our review of the 76 papers with primary endpoint changes revealed that none provided explanations for the changes to the primary outcomes. The lack of explanations for primary outcome changes does not necessarily imply issues with the study but highlights a gap in procedural transparency. Without these explanations, it is challenging to assess the appropriateness and rationale behind the changes. Therefore, while our findings suggest a higher rate of discrepancies in primary outcome reporting, it is crucial to approach these results with caution. In addition, whether certain endpoints are more prone to discrepancies was not the focus of this study and warrants further investigation in future research.

Moreover, most baseline characteristics showed no significant influence on changes in primary outcomes, except for the year of publication (year 2020 compared to 2021). The risk of discrepancy in primary outcomes remains consistent among the articles, regardless of whether they are published in high quartile journals, journals requiring compulsory registration, conducted in developed regions or multiple centers, funded by industry, prospectively registered, CONSORT endorsed, etc. This discovery aligns with a study by Damen et al. [28], which scrutinized 163,129 RCTs and identified a modification rate of 22.1% in primary outcomes. However, specific attributes of the author's team were the only factors associated with a reduced risk of modifications in primary outcomes. Our study diverges from prior research by focusing

on top journals within different quartiles. Our initial hypothesis was that top journals in the higher-ranked quartile would exhibit fewer discrepancies in trial reporting. Contrary to expectations, we found no significant differences in the risk of outcome reporting bias based on the journal's impact factor quartile, suggesting that quartile may not reliably indicate the reporting consistency of primary outcomes. Interestingly, we found no correlation between the mention of adherence to CONSORT guidelines and the mandatory nature of registration with discrepancies in primary outcomes. Simply stating compliance with CONSORT in publications does not guarantee strict adherence to all recommendations [27]. Researchers may tend to report positive results while neglecting non-significant ones, as evidenced by our findings revealing that 57.1% of prospectively registered studies with primary outcome discrepancies tended to report statistically significant results and evidenced by another article indicating a primary result spin of 66.6% in RCTs of endometriosis pain [33]. This bias could be attributed to the traditional preference for publishing studies with positive outcomes over those with negative ones [34, 35]. Despite some journals requiring a complete CONSORT checklist during manuscript submission or recommending reporting guidelines, the extent to which journals or peer reviewers verify authors' compliance with all checklist items remains unclear. Potential factors may include insufficient training resources for peer reviewers, with only a 15% training rate for reviewing clinical trials [36]. Concerning mandatory registration requirements, certain journals may announce compulsory registration of clinical trials, but editors may sometimes overlook registration requirements and exhibit a lack of scrutiny in enforcing trial registration.

## Strengths and limitations

**Strengths.** Our study included journals across impact factor quartiles, not limited to high impact factor journals, to evaluate reporting issues, thus enhancing the generalizability of the findings. The study's contemporary 5-year timespan increases the sample's generalizability and relevance to current research practices. In addition, we include as many basic features of the study as possible to ensure a thorough examination of factors influencing discrepancies.

**Limitations.** Firstly, our research only evaluated modifications to primary outcomes, as these are crucial for addressing the primary research question. Selective reporting bias could be reflected through modifications in other trial design elements after registration, e.g., secondary outcomes and sample size, which we did not inspect. Secondly, the lack of blinding researchers to the registration status may introduce bias in data extraction and analysis. Thirdly, this study is restricted to a specific field and covers a relatively small period of time, specifically five years. Consequently, the findings of this study may not be generalizable to other fields or longer timeframes. Further research is needed to explore these aspects in different contexts and over extended periods. Finally, although we conducted a thorough search to determine whether a study was registered, we did not assess whether the study protocols were published prior to the registration of the trials. It is possible that some trials were not registered due to the existence of a prior published protocol. Future research could benefit from investigating this aspect.

## Implications and actions needed

Firstly, the prevalence of discrepancies within gastroenterology and hepatology journals, particularly those favoring statistically significant primary outcomes, underscores a broader issue within the scientific community—a prevailing bias against negative or non-significant results. Therefore, there is a pressing need for a cultural shift that embraces and values negative findings. Sterling highlighted in 1959 that studies with significant findings were more likely to be

published than those with non-significant results [37]. Meanwhile, evidence also shows a gradual shift. A study in 2017 covering publication trends from 1985 to 2013 found that while significant results still dominate, there is a notable increase in the reporting of non-significant results across various journals, suggesting a slow but positive change in attitudes towards these findings [38]. This proactive approach would probably contribute to mitigating reporting bias.

Secondly, it is imperative to provide comprehensive training for authors. For instance, emphasizing the importance of trial registration, updating registries when primary outcomes are modified, and transparently reporting both positive and negative results. Moreover, training should extend beyond authors to include biostatisticians, clinical research coordinators, and staff at clinical study centers. This comprehensive training approach ensures that all parties involved in clinical trials are aware of the standards necessary for transparent reporting.

Thirdly, the observed shortcomings in the quality of RCT reviews suggest that raising the awareness among editors and reviewers involved in assessing RCTs to check registration information is crucial. Editors could receive training to strengthen their capabilities to rigorously evaluate trial protocols, registration details, and outcome reporting. One underlying issue is inadequate training sources and content, a key issue which requires attention [36]. Reviewers should work closely with editors to ensure consistency between registration and reporting in manuscripts, and make more specialized comments when needed.

Finally, more journals should explicitly outline registration requirements in their instructions for authors, avoiding vague statements such as "encouragement of trial registration." Furthermore, while some journals require mandatory registration in the author's instruction, they still publish studies without registration, indicating oversight in the review process. Implementing a more robust checking mechanism in the editor workflow, such as demanding registration identifiers before peer review, reviewing any revisions by comparing the manuscript with versions of the registered protocol, and giving special attention to the timing of primary outcomes, as well as any omissions or introductions of primary outcomes, could significantly improve authors' motivation to adhere to registration, update, and report any changes. Additionally, implementing artificial or artificial intelligence-assisted review on the transparency and completeness of clinical trial reporting following the CONSORT checklist could further promote transparency and reliability in clinical trial reporting.

## Conclusions

Our findings reveal that despite increasing registration rates, inconsistency between the pre-registered primary outcomes and those reported in publications persists. Notably, factors such as high journal quartile, developed regions, industry funding, multiple centers, prospectively registered trials, adherence to CONSORT guidelines, and mandatory registration did not show a significant association with discrepancies in primary outcome reporting. This lack of association across various trial characteristics suggests that discrepancies in primary outcome reporting are not confined to specific types of trials or contexts, but rather, are a prevalent issue within the gastroenterology and hepatology research community. Therefore, detailed registration and updates of primary and secondary outcomes in trial registries are heavily warranted. Efforts to improve reporting practices are recommended to be driven by journal-level policies and workflows. This includes implementing measurements such as withholding support for clinical trials that do not disclose registry information or do not report discrepancies between pre-specified and reported outcomes. Editorial checking mechanisms should also be put in place to ensure the transparency and reliability of reported trial outcomes.

## Supporting information

**S1 File. Search strategy.**
(DOCX)

**S2 File. STROBE checklist.**
(DOCX)

**S3 File. Data extraction form.**
(DOCX)

**S1 Appendix. Dataset.**
(XLSX)

## Acknowledgments

We acknowledge the language polish by Brad Li.

## Author Contributions

**Conceptualization:** Bing-Han Shang, Kai-Ping Zhang.

**Data curation:** Bing-Han Shang, Fang-Hui Yang, Yao Lin, Kai-Ping Zhang.

**Formal analysis:** Bing-Han Shang, Kai-Ping Zhang.

**Methodology:** Bing-Han Shang, Kai-Ping Zhang.

**Supervision:** Kai-Ping Zhang.

**Validation:** Bing-Han Shang.

**Writing – original draft:** Bing-Han Shang, Kai-Ping Zhang.

**Writing – review & editing:** Bing-Han Shang, Fang-Hui Yang, Yao Lin, Szymon Bialka, Dina Christa Janse van Rensburg, Adriano R. Tonelli, Sheikh Mohammed Shariful Islam, Izumi Kawagoe, Caroline Rhéaume, Kai-Ping Zhang.

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
