## [Decision Letter · Decision Letter 0]

15 Aug 2024

PONE-D-24-20434Discrepancies between pre-specified and reported primary outcomes: a cross-sectional analysis of randomized controlled trials in gastroenterology and hepatology journalsPLOS ONE

Dear Dr. Zhang,

Thank you for submitting your manuscript to PLOS ONE. After careful consideration, we feel that it has merit but does not fully meet PLOS ONE’s publication criteria as it currently stands. Therefore, we invite you to submit a revised version of the manuscript that addresses the points raised during the review process.

We look forward to receiving your revised manuscript.

Kind regards,

Sascha Köpke

Academic Editor

PLOS ONE

Journal Requirements:

2. Thank you for stating the following in the Competing Interests section: "I have read the journal's policy and the authors of this manuscript have the following competing interests: Adriano R.  Tonelli plays as an advisory board for Janssen and Merck and received grant from Janssen; Szymon Bialka is the secretary of the Silesian Branch of the Polish Society of Anesthesiology and Intensive Care from 2021 till now, the president of the Section of Regional Anesthesia and Pain Therapy and Polish Society of Anesthesiology and Intensive Care from 2021 til now, continued Treasurer of the Polish Society of Regional Anesthesia and Pain Therapy from 2023 till now, a co-creator patent device for isolating a patient with suspected infectious disease (exclusive right number: Pat.243051)."

Reviewers' comments:

Reviewer's Responses to Questions

**Comments to the Author**

1. Is the manuscript technically sound, and do the data support the conclusions?

Reviewer #1: Yes

Reviewer #2: Yes

2. Has the statistical analysis been performed appropriately and rigorously? 

Reviewer #1: I Don't Know

Reviewer #2: Yes

3. Have the authors made all data underlying the findings in their manuscript fully available?

Reviewer #1: Yes

Reviewer #2: Yes

4. Is the manuscript presented in an intelligible fashion and written in standard English?

Reviewer #1: Yes

Reviewer #2: No

5. Review Comments to the Author

Reviewer #1: Dear editor,

dear authors,

thank you for giving me the opportunity to review the paper entitled „Discrepancies between pre-specified and reported primary outcomes: a cross-sectional analysis of randomized controlled trials in gastroenterology and hepatology journals“.

The paper describes registration rates, discrepancies between pre-specified and reported primary outcomes, as well as publication and reporting bias in RCTs focusing on journals in gastroenterology and hepatology. While registration rates are found to be relatively high and stable over time, about one quarter of assessed trials contain discrepancies in their registered and reported primary outcomes, in particular differences in assessment timing. The manuscript also reports many statistical tests aimed at finding associations of these practices with trial or journal characteristics.

I would like to congratulate the authors to their well-conducted work that adds important knowledge on registration rates and outcome switching in a specific field.

*****

During review, I fould a few major points that I would suggest the authors address:

First, I would like to suggest that you have a native English speaker read through the document. I am not a native speaker myself, but I got the impression that there are a few sections that could use a bit of language polishing.

Second, could you provide - if available - any link of reference to where your study was preregistered?

ll. 28-29:

I think the statement „However, whether similar discrepancies exist in RCTs focusing on gastrointestinal and liver diseases remains unclear.“ is not warranted. You yourself state that another study has already been conducted in the field. Please tone down that statement.

ll. 45.46 (and similar text passages throughout the manuscript):

I consider sentences like "There was a lower risk of primary outcome discrepancies associated with more recent publication years (OR=0.267, 46 95% CI: 0.101, 0.706, p=0.008)" in the Abstract's conclusions a bit overstretching. You report a large number of univariate analyses (I counted approximately 50). Under the circumstances, this could as well be a chance finding and should not be featured so prominently.

ll. 114-118:

I would like to better understand the reasoning how the journals were chosen. Why did you choose the "top" journals in each quartile, as oppopsed to, for example, a random sample from each quartile or a random sample overall? I am sure you have very good reasons for your choice, but more detail would help readers understand this better.

l. 166:

Sentence „or if a nonsignificant primary outcome was omitted“. I am still struggling to understand how you can assess this. If the outcome was omitted, there is no way of telling whether it was significant or non-significant, right? Please explain this in a bit more detail.

Table 3:

I don’t think this Table works as it is. Although it is very concise (which was probably your intention), I think it will be rather confusing to readers because the numbers do not add up. My suggestion is that you make three different tables, with:

Table 3a - Column 1: Registered trials, Column 2: Unregistered trials, Column 3: Total of all trials.

Table 3b - Column 1: Trials with prospective registration, Column 2: Trials with retrospective registration, Column 3: Total of all registered trials.

Table 3c - Column 1: Trials with primary outcome agreement, Column 2: Trials with primary outcome disagreement, Column 3: Total of all eligible trials (i.e., all trials for which primary outcome discrepancies were assessed).

Then you can present the P-Values at the right side (similar to Table 4), so it becomes clear which groups were compared. I know this leads to more and larger tables, but I think they will be much more understandable. Of course, feel free to choose a different presentation if you have a better idea.

Figure 1:

The replacement of journals is not included in the flowchart. If one took it literally, one would expect that only trials from 10 journals were included.

Figure 2A:

For the Table on the right, the year should rather be the top row, not the bottom row.

*****

Please also note the following minor points:

Short title:

The short title reads „Discrepancies between registered and reported primary outcome“ - it probably should be „primary outcomes“?

l. 38:

I suggest you do not list all the statistical tests and models used, but instead, only the tests that are reported in the Abstract’s results section (which I understand is only the logistic regression model).

l. 45 (and other parts in the text):

Please rather write „lower chance“ instead of „lower risk“. I think "risk" has too much of an accusing tone in this context.

ll. 47-48:

Regarding the non-significant influences, please also consistently report ORs and CIs.

ll. 77-79:

Please provide the original citation for this finding, not another paper that cited the original source. If possible, I’d also suggest you report a median instead of a mean, with no decimal places.

ll.129-130:

I don’t understand why you did not include crossover trials? Are they not a type of randomized-controlled trial? (This could be a misunderstanding from my side, of course.)

ll. 130-131:

Can you provide more detail on how exactly you searched for the full text?

l. 137:

Rather „resolved through discussion“ instead of „resolved through consensus“?

ll. 147-148:

„If the registry number was uncovered in this way, the trial was deemed unregistered.“ Can you provide a reasoning why?

ll. 169-170:

How are „initially registered“ primary outcomes defined? As the first registry entry at all? Or the first registry entry after study start? Did all of the included registries actually allow for an assessment of historical versions? Please provide more detail on these questions.

ll. 193-196:

Minor suggestion: Since the journal names are already in the table, I’d leave them out of the text.

Table 5:

Please write „univariate analyses“ (plural, not singular).

l. 292 („28% to 88.4%“):

Please be consistent in how many digits after the comma you report.

ll. 388-389:

„Notably, factors such as high journal quartile, developed regions, industry funding, multiple centers, prospectively registered trials, adherence to CONSORT guidelines, and mandatory registration did not show a significant association with discrepancies in primary outcome reporting. This suggests that discrepancies in primary outcome reporting may be a prevalent issue within the gastroenterology and hepatology research community.“ I dont understand how the second sentence follows the first logically.

Again, thank you for letting me review this paper, and good luck with the publication process!

Reviewer #2: First of all, the authors have to be congratulated for this interesting and extensive study. However, there are some points that need to revised, especially regarding the presentation of methods and of the results in the tables.

1. Abstract: Please include in the methods from where the information on prespecified primary endpoints was taken.

2. Please specify the “more recent publication” year in the results section.

3. Introduction: Generally, the introduction is well written, but on line 97 there seem to be break when introducing the field of gastroenterology. Why should there be a study in this field, when we already have data across other fields? Why should findings differ in gastroenterology?

4. Materials & methods: Please delete the statement regarding PRISMA (line 118) as PRISMA is, firstly, a reporting guideline and, secondly, only suitable for systematic reviews.

5. The screening process seems a bit unclear. Did 3 investigators independently screen each title/ abstract as well as full-text? Why did you choose 3 instead of 2?

6. I was wondering whether discrepancies regarding endpoint assessment were not an own category, for instance, when instead of 2 independent investigators to assess an endpoint data from patients’ records were drawn or quality of life was measured using another instrument than prespecified in the protocol.

7. Results: You write that figure 2 shows the trends (line 228), but there are no trends over the years. The figure just shows the distribution over years.

8. I have some difficulties in understanding table 3. For instance, you write that there are differences in study location between registered and unregistered trials. However, study locations of unregistered trials are not shown in table 3. This is not only confusing for the reader (and unclear to which comparisons these p-values belong to) but also omits important results. The same is true for all other columns, as one wants to compare studies with vs. those without prospective registration, but data are not shown.

9. You further have to think whether these proportions presented in table 3 are really necessary or whether the important information is which proportion of trials (e.g. from 2017, 2018 … 2021 or by study location) are registered, prospective registered and showed agreement between registry and publication. In my opinion, these are the very informative results here that are important to know for the reader. This is also true for totals, the important information is not that the crude denominator with agreements is 209 but that the proportion is 209/285=73.3%. Maybe you can also show both proportions (and maybe it would also be more appropriate not to show proportions with (dis)agreement here in table 3 but in table 5, see my comment below).

10. Table 5 is also a difficult one. You show that in 2021 a statistically significant lower proportion of disagreements was found. But, again, the proportion of (dis)agreements by years (or any other covariates) is never shown. What you are showing here is not the information the reader needs. Overall, tables 3 and 5 need to be rearranged and more stratified proportions should also be described in the text.

11. Furthermore, in the header of table 5, odds ratios from the logistic regression should also be named as such (as you also showed kappa coefficients, which would otherwise be confusing for the reader).

12. Discussion: A (although weak) reason for not registering a trial maybe that a study protocol was published beforehand. Did you assess this?

13. Furthermore, to get it right, what about studies that did not report registration (or a registration number) but were indeed registered. Have you checked that possibility and how often was that the case? Maybe this also needs to be more elaborated in methods sections as it is not clear how you assessed whether a study was registered or not. Along with that, you write that you found invalid registration numbers (line 299). How did you deal with that (was this a non-registered study)?

14. One important piece of information would also be which endpoints were most often changed (e.g. mortality or quality of life) and in which studies (e.g. on drugs effectiveness or screening). Was this assessed or is there some kind of data from the literature on that?

15. On line 310-313, you discuss that your findings of higher proportions of disagreements than in earlier studies might imply that a divergence has concurrently risen. However, this is a bad argument not supported by your data, where in 2021 this proportions was even lower. Are there any other methodological explanations?

16. Furthermore, there might be (more or less plausible) reasons to change endpoints, that, of course, need to be transparently discussed in a study publication. Have you assessed how often this was the case (as you write in line 320 that in many cases this was not explained)? When you have assessed this systematically, these data need to be presented. Otherwise, you need to be somewhat more cautious in your own discussion.

17. I do not understand what you mean with the inabilities in the registry (lines 357-359).

18. Other limitations include the restrictions of your work to one field and a small period of time.

19. In my opinion, implications and actions needed (lines 361-382) are some kind of superficial but at the same time also too enthusiastic. Is there really a need for a cultural shift (this is a very strong statement)? Is there evidence that more and more journals recognize non-significant results (and since when is this the case)?

20. Is there really a need for training of reviewers (lines 369-372)? Is it the task of a reviewer to check whether endpoints were changed? In the following paragraph you write, that this should be done by the journal.

21. What about authors? They need to report such changes transparently. This should be trained in courses on good clinical practice and biometricians or clinical study centers need also be aware of this (as they often supervising many RCTs).

6. PLOS authors have the option to publish the peer review history of their article (what does this mean?). If published, this will include your full peer review and any attached files.

Reviewer #1: **Yes: **Martin Holst

Reviewer #2: No

---

## [Author Response · Author response to Decision Letter 0]

9 Sep 2024

ID: PONE-S-24-26486

Manuscript: Discrepancies between pre-specified and reported primary outcomes: a cross-sectional analysis of randomized controlled trials in gastroenterology and hepatology journals

Dear Academic Editor Sascha Köpke, Dr. Martin Holst (Reviewer #1) and Reviewer #2,

We are very grateful to you for the constructive comments on our paper, which have substantially improved the overall quality of this manuscript. All the comments have been carefully reviewed and addressed, with corresponding changes denoted in the manuscript. A clean, unannotated copy of the revised paper is also provided for your review. Additionally, we have provided further explanations where needed. We believe these revisions have strengthened the manuscript and made it more informative for our readers. Once again, thank you for your invaluable input. We look forward to your feedback on the revised manuscript.

Best Regards,

Kaiping Zhang, PhD, MPH

Point-to-point Reply to Reviewers’ Comments

Reviewer #1

Dear editor, dear authors,

Thank you for giving me the opportunity to review the paper entitled “Discrepancies between pre-specified and reported primary outcomes: a cross-sectional analysis of randomized controlled trials in gastroenterology and hepatology journals”.

The paper describes registration rates, discrepancies between pre-specified and reported primary outcomes, as well as publication and reporting bias in RCTs focusing on journals in gastroenterology and hepatology. While registration rates are found to be relatively high and stable over time, about one quarter of assessed trials contain discrepancies in their registered and reported primary outcomes, in particular differences in assessment timing. The manuscript also reports many statistical tests aimed at finding associations of these practices with trial or journal characteristics.

I would like to congratulate the authors to their well-conducted work that adds important knowledge on registration rates and outcome switching in a specific field.

Reply: Thank you for your positive comments and effort in reviewing our manuscript. 

During review, I found a few major points that I would suggest the authors address:

Comment 1

First, I would like to suggest that you have a native English speaker read through the document. I am not a native speaker myself, but I got the impression that there are a few sections that could use a bit of language polishing.

Reply: The manuscript was proofread by three native English speakers before its submission. However, we understand that some language issues may still exist and bring potential barriers to understanding. The manuscript is now thoroughly polished by a native English speaker.

Changes in the text: Please see the revised manuscript. 

Comment 2

Second, could you provide - if available - any link of reference to where your study was preregistered?

Reply: Thank you for your comment. We did not preregister for the study. We thought the study could be waived of mandatory registration, considering that it is a retrospective and observational study (not a clinical trial), and that the study is only based on published and open records (without retrieving any patients’ data). However, if the reviewer feels that registration is required, we would be happy to conduct a supplementary registration.

Changes in the text: None.

Comment 3

ll. 28-29:

I think the statement “However, whether similar discrepancies exist in RCTs focusing on gastrointestinal and liver diseases remains unclear” is not warranted. You yourself state that another study has already been conducted in the field. Please tone down that statement.

Reply: Many thanks for the suggestion. We have adjusted the statement accordingly. 

Changes in the text: Lines 28-29 "However, studies examining whether similar discrepancies exist in RCTs focusing on gastrointestinal and liver diseases are limited ". 

Comment 4

ll. 45.46 (and similar text passages throughout the manuscript):

I consider sentences like "There was a lower risk of primary outcome discrepancies associated with more recent publication years (OR=0.267, 46 95% CI: 0.101, 0.706, p=0.008)" in the Abstract's conclusions a bit overstretching. You report a large number of univariate analyses (I counted approximately 50). Under the circumstances, this could as well be a chance finding and should not be featured so prominently.

Reply: We appreciate your valuable suggestion. We apologize for any confusion. The univariate analysis included 10 variables, with the only significant difference finding related to the publication year 2021. We agree that this could be a chance finding and have revised the sentence to reflect that possibility, as well as specifying the exact year ‘2021’. 

Changes in the text: Lines 47 "There was a lower chance of primary outcome discrepancies associated with more recent publication year (year 2021) (OR=0.267, 95% CI: 0.101, 0.706, p=0.008)."

Comment 5

ll. 114-118:

I would like to better understand the reasoning how the journals were chosen. Why did you choose the "top" journals in each quartile, as oppopsed to, for example, a random sample from each quartile or a random sample overall? I am sure you have very good reasons for your choice, but more detail would help readers understand this better. 

Reply: Thank you for your insightful comment. We agree that including journals via a random sample could provide a broader perspective. However, we selected the "top" journals from each quartile to better address our initial research hypothesis and question. We initially hypothesized that RCTs published in higher-ranked journals may demonstrate better consistency in primary outcome reporting, so we stratified our selection by quartile. Additionally, we aimed to assess primary outcome consistency specifically within the “top” journals of each quartile. Thus, the "top" concept encompasses both between and within quartile considerations. Interestingly, as shown in table 5, we found no significant differences between different quartiles of journals. We acknowledge your point and have provided further details for clarity. 

Changes in the text: Lines 343-347 "Our study diverges from prior research by focusing on top journals within different quartiles. Our initial hypothesis was that top journals in the higher-ranked quartile would exhibit fewer discrepancies in trial reporting. Contrary to expectations, we found no significant differences in the risk of outcome reporting bias based on the journal’s impact-factor quartile, suggesting that quartile may not reliably indicate reporting consistency".

Comment 6

l. 166:

Sentence "or if a nonsignificant primary outcome was omitted”. I am still struggling to understand how you can assess this. If the outcome was omitted, there is no way of telling whether it was significant or non-significant, right? Please explain this in a bit more detail.

Reply: Thank you for your insightful question. You are correct that if an outcome is omitted, we cannot directly determine whether it was significant or non-significant. Therefore, we followed the approach outlined by Chan AW et al. (JAMA 2004;291(20):2457-2465. PMID: 15161896), where an omitted primary outcome is defined as one listed in the registration but missing from the published article with no reported results. A reasonable assumption is that the omitted outcome might not have achieved statistical significance, leading to its exclusion from the published results. This assumption is based on the notion that statistically significant results are more likely to be reported and published due to the well-known publication bias. Regretfully, 

no one else can directly assess its significance (except the authors themselves). We hope this clarifies our methodology.

Changes in the text: None.

Comment 7

Table 3: I don’t think this Table works as it is. Although it is very concise (which was probably your intention), I think it will be rather confusing to readers because the numbers do not add up. My suggestion is that you make three different tables, with: Table 3a - Column 1: Registered trials, Column 2: Unregistered trials, Column 3: Total of all trials. Table 3b - Column 1: Trials with prospective registration, Column 2: Trials with retrospective registration, Column 3: Total of all registered trials. Table 3c - Column 1: Trials with primary outcome agreement, Column 2: Trials with primary outcome disagreement, Column 3: Total of all eligible trials (i.e., all trials for which primary outcome discrepancies were assessed). Then you can present the P-Values at the right side (similar to Table 4), so it becomes clear which groups were compared. I know this leads to more and larger tables, but I think they will be much more understandable. Of course, feel free to choose a different presentation if you have a better idea.

Reply: We appreciate this informative comment and have revised Table 3 as suggested.

Changes in the text: Please see the revised Table 3.

Table 3. Registration, prospective registration, and primary outcome agreement across clinical trials, stratified by trial characteristics

Table 3a

Feature Total (n=362) Registered trials (n=312) Unregistered trials(n=50) P-Value

Year of publication, n (%) 0.83

2017 73 63(86.3) 10(13.7) 

2018 64 52(81.3) 12(18.7) 

2019 82 76(92.7) 6(7.3) 

2020 75 61(81.3) 14(18.7) 

2021 68 60(88.2) 8(11.8) 

Study Location, n (%) 

Europe 148 131(88.5) 17(11.5) 0.009 *

America 100 91(91.0) 9(9.0) 

Asia 99 78(78.8) 21(21.2) 

Oceania 9 9(100.0) 0(0.0) 

Africa 6 3(50.0) 3(50.0) 

Quartile, n (%) <0.001*

Q1 197 185(93.9) 12(6.1) 

Q2 52 39(75.0) 13(25.0) 

Q3 50 39(78.0) 11(22.0) 

Q4 63 49(77.8) 14(22.2) 

Funding, n (%) 0.01*

Industry 109 97(89.0) 12(11.0) 

Nonprofit 156 139(89.1) 17(10.9) 

Mixed 17 16(94.1) 1(5.9) 

None disclosed/no funding 80 60(75.0) 20(25.0) 

Trial design, n (%) 0.21

Superiority 338 289(85.5) 49(14.5) 

Equivalency 4 4(100.0) 0(0.0) 

Noninferiority 20 19(95.0) 1(5.0) 

Table 3b

Feature Total (n=312) Prospective Registration (n=249) Retrospective Registration (n=63) P-value

Year of publication, n (%) 0.92

2017 63 51(81.0) 12(19.0) 

2018 52 42(80.8) 10(19.2) 

2019 76 57(75.0) 19(25.0) 

2020 61 53(86.9) 8(13.1) 

2021 60 46(76.7) 14(23.3) 

Study Location, n (%) 0.74

Europe 131 105(80.2) 26(19.8) 

America 91 76(83.5) 15(16.5) 

Asia 78 59(75.6) 19(24.4) 

Oceania 9 7(77.8) 2(22.2) 

Africa 3 2(66.7) 1(33.3) 

Quartile, n (%) <0.001*

Q1 185 163(88.1) 22(11.9) 

Q2 39 25(64.1) 14(35.9) 

Q3 39 31(79.5) 8(20.5) 

Q4 49 30(61.2) 19(38.8) 

Funding, n (%) 0.001*

Industry 97 90(92.8) 7(7.2) 

Nonprofit 139 102(73.4) 37(26.6) 

Mixed 16 14(87.5) 2(12.5) 

None disclosed/no funding 60 43(71.7) 17(28.3) 

Trial design, n (%) 0.006*

Superiority 289 230(79.6) 59(20.4) 

Equivalency 4 1(25.0) 3(75.0) 

Noninferiority 19 18(94.7) 1(5.3) 

Table 3c

Feature Total (n=285) Primary-Outcome Agreement Between Registry and Publication (n=209) Primary-Outcome Disagreement Between Registry and Publication (n=76) P-value

Year of publication, n (%) 0.53

2017 56 37(66.1) 19(33.9) 

2018 50 39(78.0) 11(22.0) 

2019 68 47(69.1) 21(30.9) 

2020 58 51(87.9) 7(12.1) 

2021 53 35(66.0) 18(34.0) 

Study Location, n (%) 0.51

Europe 120 85(70.8) 35(29.2) 

America 82 65(79.3) 17(20.7) 

Asia 72 50(69.4) 22(30.6) 

Oceania 8 7(87.5) 1(12.5) 

Africa 3 2(66.7) 1(33.3) 

Quartile, n (%) 0.17

Q1 174 134(77.0) 40(23.0) 

Q2 34 21(61.8) 13(38.2) 

Q3 36 27(75.0) 9(25.0) 

Q4 41 27(65.9) 14(34.1) 

Funding, n (%) 0.18

Industry 94 76(80.9) 18(19.1) 

Nonprofit 128 88(68.8) 40(31.2) 

Mixed 11 9(81.8) 2(18.2) 

None disclosed/no funding 52 36(69.2) 16(30.8) 

Trial design, n (%) 0.29

Superiority 264 191(72.3) 73(27.7) 

Equivalency 3 3(100.0) 0(0.0) 

Noninferiority 18 15(83.3) 3(16.7) 

Q: Quartile.

*: p< 0.05.

Comment 8

Figure 1: The replacement of journals is not included in the flowchart. If one took it literally, one would expect that only trials from 10 journals were included.

Reply: Thank you for this important comment. We have now included the process of replacing the two journals.

Changes in the text: Please see revised Figure 1.

Comment 9

Figure 2A:

For the Table on the right, the year should rather be the top row, not the bottom row. 

Reply: Many thanks for the suggestion. Considering that the revised Table 3 contains repeated information in the previous Figure 2, we now have deleted Figure 2. 

Changes in the text: Figure 2 is deleted. Please see the revised Table 3. 

Comment 10

Short title: The short title reads “Discrepancies between registered and reported primary outcome” - it probably should be „primary outcomes ?

Reply: We apologize for this mistake and have changed it. 

Changes in the text: Short title: "Discrepancies Between Registered and Reported Primary Outcomes"

Comment 11

l. 38: I suggest you do not list all the statistical tests and models used, but instead, only the tests that are reported in the Abstract’s results section (which I understand is only the logistic regression model).

Reply: Thank you for your valuable feedback. We understand your suggestion to list only the statistical tests and models reported in the Abstract's results section. However, we believe it is important to mention all statistical analysis methods in the Abstract's methods section to provide a complete overview of our methodology used. Therefore, we have decided to retain this information to ensures clarity and comprehensiveness for readers.

Changes in the text: None. 

Comment 12

l. 45: (and other parts in the text): Please rather write "lower chance “instead of "lower risk”. I think "risk" has too much of an accusing tone in this context.

Reply: Thank you for your advice. We have reviewed the full text and made revisions accordingly.

Changes in the text: Line 47 "There was a lower chance of primary outcome discrepancies associated with more recent publication years (Year 2021) (OR=0.267, 95% CI: 0.101, 0.706, p=0.008)". And Lines 288-289 "A more recent publication year was associated with a lower chance of primary outcome discrepancy".

Comment 13

ll. 47-48: Regarding the non-significant influences, please also consistently report ORs and CIs.

Reply: Many thanks for the suggestion. We conducted the chi-square test to assess the relationship between journal quartiles and both primary outcome consistency and potential reporting bias. We want to clarify that in the context of chi-square tests, it is standard practice to report p-values to indicate statistical significance. Including ORs and CIs for non-significant results may not provide additional meaningful information. We hope this clarifies our reporting approach. 

Changes in the text: None.

Comment 14

ll. 77-79A: Please provide the original citation for this finding, not another paper that cited the original source. If possible, I’d also suggest you report a median instead of a mean, with no decimal places. 

Reply: Thank you for your insightful feedback. We acknowledge the importance of citing the original source and revised our citation accordingly. Regarding the reporting of median values, we reviewed the original source and found that only mean value were reported, specifically "On average, each trial reported just 58.2% of its specified outcomes. And on average, each trial silently added 5.3 new outcomes." Therefore, we have retained the mean values in our article. 

Changes in the text: Reference 6: Goldacre B, Drysdale H, Smith AP, Dale A, Milosevic A, Slade E, et al. The COMPare Trials project. Available from: https://www.compare-trials.org/

Comment 15

ll.129-130: I don’t understand why you did not include crossover trials? Are they not a type of randomized-controlled trial? (This could be a misunderstanding from my side, of course.) 

Reply: Thank you for your thoughtful feedback. We acknowledge that crossover trials are indeed a type of RCT. However, the primary objective of our study was to evaluate the consistency between registered and published outcomes in parallel-group RCTs comparing two different treatment modalities. While valuable, crossover trials involve a different design where the same participants receive 

---

## [Decision Letter · Decision Letter 1]

17 Oct 2024

PONE-D-24-20434R1Discrepancies between pre-specified and reported primary outcomes: a cross-sectional analysis of randomized controlled trials in gastroenterology and hepatology journalsPLOS ONE

Dear Dr. Zhang,

Thank you for submitting your manuscript to PLOS ONE. After careful consideration, we feel that it has merit but does not fully meet PLOS ONE’s publication criteria as it currently stands. Therefore, we invite you to submit a revised version of the manuscript that addresses the points raised during the review process. The reviewers have provided some further comments that should be considered for a possible revision. In addition, I would ask you to reconsider your answer to comment (13) by reviewer 1 and include ORs and CIs also for non-significant results (or provide a more convincing explanation for not doing so).

Kind regards,

Sascha Köpke

Academic Editor

PLOS ONE

Journal Requirements:

Reviewers' comments:

Reviewer's Responses to Questions

**Comments to the Author**

1. If the authors have adequately addressed your comments raised in a previous round of review and you feel that this manuscript is now acceptable for publication, you may indicate that here to bypass the “Comments to the Author” section, enter your conflict of interest statement in the “Confidential to Editor” section, and submit your "Accept" recommendation.

Reviewer #1: All comments have been addressed

Reviewer #2: All comments have been addressed

2. Is the manuscript technically sound, and do the data support the conclusions?

Reviewer #1: Yes

Reviewer #2: Partly

3. Has the statistical analysis been performed appropriately and rigorously? 

Reviewer #1: Yes

Reviewer #2: No

4. Have the authors made all data underlying the findings in their manuscript fully available?

Reviewer #1: Yes

Reviewer #2: Yes

5. Is the manuscript presented in an intelligible fashion and written in standard English?

Reviewer #1: Yes

Reviewer #2: Yes

6. Review Comments to the Author

Reviewer #1: Dear authors,

dear editor,

thank you for the opportunity to again review the manuscript entitled „Discrepancies between pre-specified and reported primary outcomes: a cross-sectional analysis of randomized controlled trials in gastroenterology and hepatology journals.“

I would like to thank the authors for their extensive effors to implement the changes requested by the reviewers, including reworking all the tables (whic I find much more readable now!). I believe the maunscript has greatly improved and can be accepted for publication. There are, however, three minor points I would like to address:

(1) My Comment 6

You provided a detailed answer to my question, thank you! However, you made no changes to the text. I would appreciate if you could explain this to readers, too.

(2) My Comment 16

You added that there was only one paper for which you could not obtain the full text. Could you also add this information to the manuscript?

(3) My Comment 18

Thank you for providing a detailed explanation! I think this was a misunderstanding, however. I had just been wondering - should the sentence not read: „If NO registry number was uncovered in this way, the trial was deemed unregistered.“ If this is a misunderstanding from my side, please feel free to ignore the comment.

Reviewer #2: Thank you for your valuable response and for revising the manuscript. Most of my points are addressed, but there are still some things that should be improved.

1. Introduction: On line 101, it still remains unclear why a study from the field of gastroenterology is needed and why results should differ from other fields.

2. On lines 105 and 107, there is now a redundancy on studies that did not assess discrepancies on journal quartiles and high vs. low impact journals.

3. Regarding my earlier comment 13, you still did not make clear how you proceed regarding whether a study was registered or not. This is important because different scenarios are possible (e.g. no registration number is in the publication and you a) did not find or b) did find a registration or c) the registration number is wrong). Please describe your approach in the methods section in more detail.

4. On line 197, it does not make sense that a binary regression included all significant variables in the univariate analysis, as both approaches assess the same and all variables were included.

5. Table 3c is the same as table 5, which is confusing. Please delete table 3c.

6. What is the outcome for the odds ratios in table 5, agreement or disagreement?

7. Regarding the year in table 5 and in the text you write that 2021 was significant different (OR=0.267). However, the proportions with disagreements between 2017 and 2021 are the same (33.9% and 34.0%). Something is wrong here. Maybe this result is for 2020. Furthermore, the odds ratios for 2018 and 2019 also do not fit together.

8. The same seem to hold true for other analyses in table 5, e.g. the proportion from Europe and Asia are quite the same (29.2% and 30.6%) but the odds ratio is given to be 0.523. Please check all analyses carefully.

9. Regarding my comment 19, it would be nice when the arguments and references given could be discussed in the manuscript.

7. PLOS authors have the option to publish the peer review history of their article (what does this mean?). If published, this will include your full peer review and any attached files.

Reviewer #1: **Yes: **Martin Holst

Reviewer #2: No

---

## [Author Response · Author response to Decision Letter 1]

1 Nov 2024

ID: PONE-S-24-26486

Manuscript: Discrepancies between pre-specified and reported primary outcomes: a cross-sectional analysis of randomized controlled trials in gastroenterology and hepatology journals

Dear Academic Editor Sascha Köpke, Dr. Martin Holst (Reviewer #1) and Reviewer #2,

We are very grateful to you for the constructive comments on our paper, which have substantially improved the overall quality of this manuscript. All the comments have been carefully reviewed and addressed, with corresponding changes denoted in the manuscript. A clean, unannotated copy of the revised paper is also provided for your review. Additionally, we have provided further explanations where needed. We believe these revisions have strengthened the manuscript and made it more informative for our readers. Once again, thank you for your invaluable input. We look forward to your feedback on the revised manuscript.

Best Regards,

Kaiping Zhang, PhD, MPH

Point-to-point Reply to Reviewers’ Comments

Reviewer #1: 

Dear authors, dear editor,

thank you for the opportunity to again review the manuscript entitled “Discrepancies between pre-specified and reported primary outcomes: a cross-sectional analysis of randomized controlled trials in gastroenterology and hepatology journals.”

I would like to thank the authors for their extensive efforts to implement the changes requested by the reviewers, including reworking all the tables (which I find much more readable now!). I believe the manuscript has greatly improved and can be accepted for publication. There are, however, three minor points I would like to address:

Reply: Thank you for your positive feedback and efforts put in reviewing our manuscript.

Comment 1 

(1) My Comment 6

You provided a detailed answer to my question, thank you! However, you made no changes to the text. I would appreciate if you could explain this to readers, too.

Reply: Many thanks for the suggestion. We have added this in the revised manuscript.

Changes in the text: Lines 175-179 “…or if a nonsignificant primary outcome was omitted (e.g., the omitted outcome might not have achieved statistical significance, leading to its exclusion from the published results. This assumption is based on the notion that statistically significant results are more likely to be reported and published due to the well-known publication bias) or defined as nonprimary in the published articles, or if registered statistically significant secondary outcomes became published primary outcomes.”

Comment 2

(2) My Comment 16

You added that there was only one paper for which you could not obtain the full text. Could you also add this information to the manuscript?

Reply: Thank you for bringing this to our attention. We have revised it.

Changes in the text: Lines 226-228 “Subsequently, 405 articles underwent full-text screening for eligibility. Of them, 42 cross-over studies and ancillary studies were further excluded. And we were unable to obtain the full text of one trial. Finally, 362 trials were eligible for inclusion.”

Comment 3

(3) My Comment 18

Thank you for providing a detailed explanation! I think this was a misunderstanding, however. I had just been wondering - should the sentence not read: “If NO registry number was uncovered in this way, the trial was deemed unregistered.” If this is a misunderstanding from my side, please feel free to ignore the comment.

Reply: We sincerely appreciate your informative comment and apologize for this misunderstanding writing. We now have changed it.

Changes in the text: Lines 151-152 “If no registry number was uncovered in this way, the trial was deemed unregistered.”

Reviewer #2: 

Thank you for your valuable response and for revising the manuscript. Most of my points are addressed, but there are still some things that should be improved.

Reply: Thank you for your positive feedback and efforts put in reviewing our manuscript.

Comment 1 

Introduction: On line 101, it still remains unclear why a study from the field of gastroenterology is needed and why results should differ from other fields.

Reply: Thank you for your valuable feedback. We apologize for any confusion caused by our previous submission. Regarding your first concern, the burden of gastrointestinal diseases is substantial, accounting for more than one-third of prevalent disease cases globally in 2019. Despite this significant burden, research exploring the consistency between registered and published primary outcomes in the field of gastroenterology remains limited. To our knowledge, only one relevant study by Li et al. has addressed this issue within high-impact gastroenterology and hepatology journals. This scarcity of research underscores the necessity of our study to fill this gap.

Regarding your second concern, we have also reviewed literature from other biomedical fields, which has reported discrepancies between registered and published primary outcomes ranging from 5% to 92%. These variations are likely influenced not only by differences in the study design, but also by various of biomedical fields. We hope that our additional explanations and the context provided address your concerns and clarify the necessity of our study in this field.

Changes in the text: Lines 74-83 “Gastroenterology and hepatology are fields that often involve multifaceted treatment regimens and diverse patient populations. In 2019, digestive diseases accounted for more than one-third of prevalent disease cases, representing a significant global health care burden [1]. In the era of evidence-based medicine, high-quality randomized controlled trials (RCTs) stand as pivotal sources of evidence in scientific research, owing to their robust study designs and significant value [2]. These trials often serve as primary references for formulating clinical guidelines and shaping medical decision-making. However, numerous trials encounter the issue of selective and incomplete reporting of results, which distorts their evidence-based value [3-6]. The Centre for Evidence-Based Medicine Outcome Monitoring Project has found that, on average, each trial in top-ranked medical journals silently adds 5.3 new outcomes [7]. Such discrepancies in trial results can potentially exaggerate benefits or underestimate adverse outcomes, leading to misguided clinical recommendations, wastage of resources, and, in severe cases, harm to patients [8].” 

Lines 95-99, “Moreover, selective outcome reporting bias persists across biomedical disciplines, including anesthesiology, psychology, otorhinolaryngology, headache medicine, plastic surgery, mental health and orthopaedical surgery despite the implementation of ICMJE guidelines. Studies in these biomedical fields have reported a huge field difference in discrepancies between registered and published primary outcomes, ranging from 5% to 92% [13-19].”

Line 101, “Despite the huge field difference, research on this important topic remains limited in the field of gastroenterology and hepatology.”

References:

1. Wang Y, Huang Y, Chase RC, Li T, Ramai D, Li S, et al. Global Burden of Digestive Diseases: A Systematic Analysis of the Global Burden of Diseases Study, 1990 to 2019. Gastroenterology. 2023;165(3):773-83 e15. http://doi.org/10.1053/j.gastro.2023.05.050. PMID:37302558.

18. Vrljicak Davidovic N, Komic L, Mesin I, Kotarac M, Okmazic D, Franic T. Registry versus publication: discrepancy of primary outcomes and possible outcome reporting bias in child and adolescent mental health. Eur Child Adolesc Psychiatry. 2022;31(5):757-69. http://doi.org/10.1007/s00787-020-01710-5. PMID:33459886.

19. Rongen JJ, Hannink G. Comparison of Registered and Published Primary Outcomes in Randomized Controlled Trials of Orthopaedic Surgical Interventions. J Bone Joint Surg Am. 2016;98(5):403-9. http://doi.org/10.2106/JBJS.15.00400. PMID:26935463.

Comment 2

On lines 105 and 107, there is now a redundancy on studies that did not assess discrepancies on journal quartiles and high vs. low impact journals. 

Reply: Thank you for your comment. We have deleted the redundant sentence.

Changes in the text: Line 111 deleted the sentence “no recent study has examined the prevalence and nature of major outcome discrepancies in high-impact and lower-impact journals within this domain”. 

Comment 3

Regarding my earlier comment 13, you still did not make clear how you proceed regarding whether a study was registered or not. This is important because different scenarios are possible (e.g. no registration number is in the publication and you a) did not find or b) did find a registration or c) the registration number is wrong). Please describe your approach in the methods section in more detail.

Reply: Thank you for your insightful comment. We apologize for any confusion. To clarify our approach in determining whether a study was registered, we adhered to the following detailed steps: 

1) If no registration number was provided in the publication:

We manually searched the ICTRP using the publication title, author names, trial participants, and primary sponsors to identify any possible registration numbers. If no registry number was uncovered in this way, the trial was deemed unregistered. 

2) If a registration number was provided in the publication:

We entered the registration number into the ICTRP to retrieve relevant registration information. If the authors provided a registration number but we did not find a corresponding registration record, we treated these trials in the same manner as studies that did not report a registration number at all.

This comprehensive approach ensures that we thoroughly verify the registration status of each study. We hope this clarifies our methodology.

Changes in the text: Lines 149-154 “…This approach ensured uniformity in search mechanisms. If no registration number was provided in the publication, we manually searched the ICTRP using the publication title, author names, trial participants, and primary sponsors to identify any possible registration numbers. If no registry number was uncovered in this way, the trial was deemed unregistered. If a registration number was provided in the publication, we entered the registration number into the ICTRP to retrieve relevant registration information. If the authors provided a registration number but we did not find a corresponding registration record, we treated these trials in the same manner as studies that did not report a registration number at all.”

Comment 4

On line 197, it does not make sense that a binary regression included all significant variables in the univariate analysis, as both approaches assess the same and all variables were included.

Reply: Thank you for pointing out the issue. We apologize for the oversight and any confusion it may have caused. This was indeed a typographical error on our part. We have changed “binary logistic regression” to “multivariable logistic regression”. To clarify, our actual approach was to first conduct univariate analysis. Only variables that showed significant differences in the univariate analysis were then included in the multivariable logistic regression analysis. As indicated by the results in Table 5, only one independent variable was significant. Therefore, we did not proceed with multivariable regression analysis. And we also add this point in the Results section “Only the publication year was associated with differences in primary outcome reporting; therefore, multivariable logistic regression analysis was not conducted”. We appreciate your attention to detail and hope this explanation resolves the confusion. 

Changes in the text: Lines 194-197 “Additionally, univariate analyses are performed to determine the effect of each variable on primary outcome discrepancies. Multivariable logistic regression includes all the significant variables in the univariable analysis to identify factors influencing discrepancies between the registered and reported primary outcomes.

Lines 266-270 “Univariate analyses revealed that only the publication year (2020) was associated with the discrepancy between registered and published primary outcomes (OR=0.267, 95%CI:0.101,0.706, p=0.008) (Table 5); therefore, multivariable logistic regression analysis was not conducted.”

Comment 5

Table 3c is the same as table 5, which is confusing. Please delete table 3c.

Reply: Thank you for your thoughtful comment and we have deleted Table 3c. 

Changes in the text: Please see revised Table 3.

Table 3. Registration and prospective registration across clinical trials, stratified by trial characteristics.

Table 3a

Feature Total (n=362) Registered trials (n=312) Unregistered trials(n=50) P-Value

Year of publication, n (%) 0.83

2017 73 63(86.3) 10(13.7) 

2018 64 52(81.3) 12(18.7) 

2019 82 76(92.7) 6(7.3) 

2020 75 61(81.3) 14(18.7) 

2021 68 60(88.2) 8(11.8) 

Study Location, n (%) 

Europe 148 131(88.5) 17(11.5) 0.009 *

America 100 91(91.0) 9(9.0) 

Asia 99 78(78.8) 21(21.2) 

Oceania 9 9(100.0) 0(0.0) 

Africa 6 3(50.0) 3(50.0) 

Quartile, n (%) <0.001*

Q1 197 185(93.9) 12(6.1) 

Q2 52 39(75.0) 13(25.0) 

Q3 50 39(78.0) 11(22.0) 

Q4 63 49(77.8) 14(22.2) 

Funding, n (%) 0.01*

Industry 109 97(89.0) 12(11.0) 

Nonprofit 156 139(89.1) 17(10.9) 

Mixed 17 16(94.1) 1(5.9) 

None disclosed/no funding 80 60(75.0) 20(25.0) 

Trial design, n (%) 0.21

Superiority 338 289(85.5) 49(14.5) 

Equivalency 4 4(100.0) 0(0.0) 

Noninferiority 20 19(95.0) 1(5.0) 

Table 3b

Feature Total (n=312) Prospective Registration (n=249) Retrospective Registration (n=63) P-value

Year of publication, n (%) 0.92

2017 63 51(81.0) 12(19.0) 

2018 52 42(80.8) 10(19.2) 

2019 76 57(75.0) 19(25.0) 

2020 61 53(86.9) 8(13.1) 

2021 60 46(76.7) 14(23.3) 

Study Location, n (%) 0.74

Europe 131 105(80.2) 26(19.8) 

America 91 76(83.5) 15(16.5) 

Asia 78 59(75.6) 19(24.4) 

Oceania 9 7(77.8) 2(22.2) 

Africa 3 2(66.7) 1(33.3) 

Quartile, n (%) <0.001*

Q1 185 163(88.1) 22(11.9) 

Q2 39 25(64.1) 14(35.9) 

Q3 39 31(79.5) 8(20.5) 

Q4 49 30(61.2) 19(38.8) 

Funding, n (%) 0.001*

Industry 97 90(92.8) 7(7.2) 

Nonprofit 139 102(73.4) 37(26.6) 

Mixed 16 14(87.5) 2(12.5) 

None disclosed/no funding 60 43(71.7) 17(28.3) 

Trial design, n (%) 0.006*

Superiority 289 230(79.6) 59(20.4) 

Equivalency 4 1(25.0) 3(75.0) 

Noninferiority 19 18(94.7) 1(5.3) 

Q: Quartile.

*: P< 0.05.

Comment 6

What is the outcome for the odds ratios in table 5, agreement or disagreement?

Comment 7

Regarding the year in table 5 and in the text you write that 2021 was significant different (OR=0.267). However, the proportions with disagreements between 2017 and 2021 are the same (33.9% and 34.0%). Something is wrong here. Maybe this result is for 2020. Furthermore, the odds ratios for 2018 and 2019 also do not fit together.

Comment 8

The same seem to hold true for other analyses in table 5, e.g. the proportion from Europe and Asia are quite the same (29.2% and 30.6%) but the odds ratio is given to be 0.523. Please check all analyses carefully.

Reply: Thank you for your insightful comments regarding Table 5. Your comments prompted a thorough review of our analyses, leading to important corrections. Below, we address your comments in detail:

Regarding to comment 6, the odds ratios (OR) presented in Table 5 pertain to the likelihood of disagreement between registered and published primary outcomes. This means that the ORs reflect the relationship between various independent variables (e.g., publication year, quartile) and the probability of disagreement, which can indicate either protective or risk factors. An OR greater than 1 suggests a higher likelihood of discrepancy (risk factor), while an OR less than 1 suggests a lower likelihood of discrepancy (protective factor).

Regarding to comment 7, you are correct; there was an error in the initial analysis. After re-evaluating our univariate analyses, we realized that the mistakes were due to an incorrect definition of reference category, which led to inconsistencies between the OR values and the reported proportions. For example, in the revised Table 5, the OR=0.267, 95% CI (0.101, 0.706), p=0.008 indicates that the likelihood of primary outcome discrepancy in 2020 is 0.267 times that of 2021, and this difference is statist

---

## [Decision Letter · Decision Letter 2]

6 Nov 2024

PONE-D-24-20434R2Discrepancies between pre-specified and reported primary outcomes: a cross-sectional analysis of randomized controlled trials in gastroenterology and hepatology journalsPLOS ONE

Dear Dr. Zhang,

Thank you for submitting your manuscript to PLOS ONE. After careful consideration, we feel that it has merit but does not fully meet PLOS ONE’s publication criteria as it currently stands. Therefore, we invite you to submit a revised version of the manuscript that addresses the points raised during the review process.

The reviewer has highlighted further inconsistencies! So, apart from correcting the mentioned aspects, please be sure to check the complete manuscript for inconsistencies and errors.

Kind regards,

Sascha Köpke

Academic Editor

PLOS ONE

Journal Requirements:

Reviewers' comments:

Reviewer's Responses to Questions

**Comments to the Author**

1. If the authors have adequately addressed your comments raised in a previous round of review and you feel that this manuscript is now acceptable for publication, you may indicate that here to bypass the “Comments to the Author” section, enter your conflict of interest statement in the “Confidential to Editor” section, and submit your "Accept" recommendation.

Reviewer #1: All comments have been addressed

Reviewer #2: (No Response)

2. Is the manuscript technically sound, and do the data support the conclusions?

Reviewer #1: Yes

Reviewer #2: (No Response)

3. Has the statistical analysis been performed appropriately and rigorously? 

Reviewer #1: N/A

Reviewer #2: (No Response)

4. Have the authors made all data underlying the findings in their manuscript fully available?

Reviewer #1: Yes

Reviewer #2: (No Response)

5. Is the manuscript presented in an intelligible fashion and written in standard English?

Reviewer #1: Yes

Reviewer #2: (No Response)

6. Review Comments to the Author

Reviewer #1: Dear Dr Zhang, dear authors,

thank you for another opportunity to review your work and for addressing my comments. I believe they have now been completely addressed and the manuscript is acceptable for publication from my side.

Good luck and best regards,

Martin Holst

Reviewer #2: Thank you for clarifying and improving the manuscript. There are few points regarding the differences for 2020 and 2021.

1. In the abstract, in 48 please describe that the OR for the publication year 2020 is in comparison to 2021.

2. In the discussion, please correct (line 298) “A more recent publication year (year 2021) was associated with a lower chance of primary outcome discrepancy.” This result now refers to 2020 and might not be explained by some kind of trend but rather by chance.

3. This also needs to be corrected in line 345 “Moreover, most baseline characteristics showed no significant influence on changes in primary outcomes, except for the year of publication, where more recent studies were associated with fewer outcome discrepancies.”

7. PLOS authors have the option to publish the peer review history of their article (what does this mean?). If published, this will include your full peer review and any attached files.

Reviewer #1: **Yes: **Martin Holst

Reviewer #2: No

---

## [Author Response · Author response to Decision Letter 2]

7 Nov 2024

Thank you for clarifying and improving the manuscript. There are few points regarding the differences for 2020 and 2021.

1. In the abstract, in 48 please describe that the OR for the publication year 2020 is in comparison to 2021.

2. In the discussion, please correct (line 298) “A more recent publication year (year 2021) was associated with a lower chance of primary outcome discrepancy.” This result now refers to 2020 and might not be explained by some kind of trend but rather by chance.

3. This also needs to be corrected in line 345 “Moreover, most baseline characteristics showed no significant influence on changes in primary outcomes, except for the year of publication, where more recent studies were associated with fewer outcome discrepancies.”

Reply: Thank you for your further careful check and bring these inconsistencies to us. We apologize the inconsistencies due to our missed inspections after correcting the reference group. We now have revised the mentioned places and checked through the manuscript.

Changes in the text: 

1. Abstract, Lines47-48 “Univariate analyses revealed that primary outcome discrepancies were lower in the publication year 2020 compared to year 2021 (OR=0.267, 95% CI: 0.101, 0.706, p=0.008).”

2. Discussion, Lines 298-299 “Primary outcome discrepancies were lower in the publication year 2020 compared to year 2021 (OR=0.267, 95% CI: 0.101, 0.706, p=0.008).”

3. Discussion, Lines 345-346 “Moreover, most baseline characteristics showed no significant influence on changes in primary outcomes, except for the year of publication (year 2020 compared to 2021).”

4. Other revisions after checking through the manuscript are listed as below. 

--Line 40, The word “binary logistic regression” is now corrected as “logistic regression”.

--Lines 46-47 “introducing new unregistered primary outcomes” is now corrected as “reporting the registered secondary outcomes as primary outcomes”.

--Lines 49-50, “20 studies were retrospectively registered and 32 (57.1%) showed statistically significant results” is now specified as “20 (26.3%) studies were retrospectively registered, and 32 (57.1%) of the prospectively registered trials with primary outcome discrepancies showed statistically significant results.”

--Lines 98-100, “headache medicine, plastic surgery, mental health and orthopaedical surgery despite the implementation of ICMJE guidelines…. ranging from 5% to 92% [13-19]” is now corrected as “headache medicine, mental health and orthopaedical surgery despite the implementation of ICMJE guidelines… ranging from 25.9% to 92% [13-18]”.

--Line 108, “Our article addresses this gap by focusing on journals ranked in the top three of each impact factor quartile in the JCR.”, the sentence is now deleted as this information can be found in the methods.

--Table 1, the subitem of “ICMJE member” is now corrected as “ICMJE member or adherence to ICMJE guidelines”. Therefore, on Line 210, “Notably, four journals, including Liver International, Colorectal Disease, Techniques in Coloproctology, and Journal of Pediatric Gastroenterology and Nutrition, were not members of the ICMJE (Table 1)”, the sentence is now deleted. Meanwhile, all the journals regarding ICMJE member or adherence to ICMJE guidelines in table 1 are now marked as “Yes”. 

--Line 137, “Endnote” is now corrected as “Endnote (version 20; Clarivate Analytics, USA)”.

--Line 142, “Microsoft Excel” is now corrected as “Microsoft Excel (2022 version; Microsoft Corporation, USA)”.

--Line 182, “In cases where primary outcomes were added or changed, only the initially registered ones were extracted”, the sentence is now deleted as this information can be found above.

--Line 193, “The chi-square test or Fisher's exact test is used to assess differences in each trial characteristic across domains, including registration status, registration type, and primary outcome consistency” is now corrected as “The chi-square test or Fisher's exact test is used to assess differences in each trial characteristic across domains, including registration status and registration type”.

--Lines 199-200, “<0.2 indicating poor agreement” is now corrected as “≤0.2 indicating poor agreement”. “>0.81 very good agreement” is now corrected as “≥0.81 very good agreement”.

--Line 229, the sentence “while there was no difference in the number of RCTs published in Quartiles 2 to 4” is now deleted.

--Line 261, “introducing new unregistered primary outcomes” is now corrected as “registered secondary outcomes promoted to primary outcomes”.

--Line 262, “registered secondary outcomes promoted to primary outcomes in publications” is now corrected as “new primary outcomes introduced”.

--Line291, “with 68.8% registered predominantly prospectively” is now corrected as “with 86.2% (312 out of 362) registered and 79.8% (249 out of 312) registered predominantly prospectively”.

--Line 295, “introducing new unregistered primary outcomes” is now corrected as “reporting the registered secondary outcomes as primary outcomes”.

--Line 305, “with studies in psychiatry, pediatric surgery, anaesthesia, and urology reporting a range from 28.0% to 88.4% [14, 23-25]” is now corrected as “with studies in psychiatry, pediatric surgery, and anaesthesia reporting a range from 33.1% to 77.1% [14, 23, 24]”.

--Line 332-333, “omitting and introducing a primary outcome were the second and third most common discrepancies” is now corrected as “omitting a primary outcome and reporting the registered secondary outcomes as primary outcomes were the second and third most common discrepancies”.

--Line 358, the sentence is now specified as “as evidenced by our findings revealing that 57.1% of prospectively registered studies with primary outcome discrepancies tended to report statistically significant results”.

---

## [Editor Report · Decision Letter 3]

11 Nov 2024

Discrepancies between pre-specified and reported primary outcomes: a cross-sectional analysis of randomized controlled trials in gastroenterology and hepatology journals

PONE-D-24-20434R3

Dear Dr. Zhang,

We’re pleased to inform you that your manuscript has been judged scientifically suitable for publication and will be formally accepted for publication once it meets all outstanding technical requirements.

Kind regards,

Sascha Köpke

Academic Editor

PLOS ONE

---

## [Editor Report · Acceptance letter]

13 Nov 2024

PONE-D-24-20434R3 

PLOS ONE

Dear Dr. Zhang, 

I'm pleased to inform you that your manuscript has been deemed suitable for publication in PLOS ONE. Congratulations! Your manuscript is now being handed over to our production team.

Kind regards, 

on behalf of

Professor Sascha Köpke 

Academic Editor

PLOS ONE